# Rapid communication: Nonlinear sensitivity of El Niño-Southern Oscillation across climate states

Gabriel M. Pontes[1,2]*, Pedro L. Silva Dias[3] & Laurie Menviel[1,2]

**Affiliations:**

[1] Climate Change Research Centre, University of New South Wales, Sydney, NSW, Australia

[2] Australian Centre for Excellence in Antarctic Sciences, University of New South Wales, Sydney, NSW, Australia

[3] Institute of Astronomy, Geophysics and Atmospheric Sciences, University of São Paulo, São Paulo, SP, Brazil

*Corresponding author e-mail: g.pontes@unsw.edu.au.

## Abstract

The El Niño-Southern Oscillation (ENSO) is the dominant mode of tropical climate variability. Understanding its sensitivity to climate states is of societal and ecosystem importance given the unabated global warming. Paleoclimate archives and climate models suggest that ENSO activity depends on mean state conditions. However, due to climate model biases, short observational record and proxy-data uncertainties, evaluating ENSO sensitivity remains challenging. Here we combine state-of-the-art model simulations of past climates and future warming to evaluate ENSO activity throughout a wide range of climate states. We find that the sensitivity of ENSO to the background climate is nonlinear and tied to the climatological position of the tropical Pacific convection centers, namely the Intertropical and South Pacific Convergence Zones. Simulations with atmospheric $CO_2$ lower than today display a poleward shift of the convection centers and weakened ENSO. Moderate equatorward shifts of the convection centers occur under $CO_2$-induced warming increasing ENSO activity, while strong equatorward shifts reduce ENSO variability in extreme $CO_2$ warming scenarios, resulting in a permanent El Niño-like mean state. Furthermore, we find that Eastern Pacific El Niños are more sensitive to the background state than Central El Niño events. Our results provide a comprehensive mechanism of how tropical Pacific mean state modulates ENSO activity.

## 1. Introduction

ENSO is a coupled ocean-atmosphere zonal oscillation sourced in the equatorial Pacific that results in either higher (El Niño) or lower (La Niña) sea surface temperatures (SST) in the central and eastern Pacific, with strong impacts worldwide (Ropelewski & Halpert, 1987). At ~3.3 million years (Ma), greatly reduced water-mass transport through the Indonesian and Panamá seaways due to tectonic changes (De Schepper et al., 2014) likely set up conditions for the full development of ENSO, which has been the most important driver of year-to-year climate variability during the instrumental period. Before that time, during the early Pliocene (~4-5 million years ago [Ma]), it is likely the necessary conditions for ENSO development in the equatorial Pacific Ocean were not present due to zonal advection across both zonal boundaries (De Schepper et al., 2014), the absence of a strong zonal SST gradient (Fedorov et al., 2015; White & Ravelo, 2020) and a Pacific Walker cell extending to the Indian Ocean (van der Lubbe et al., 2021). While proxy-data and modelling studies provide evidence of varying ENSO activity over the last 3.3 Ma, due to changes in thermocline (Ford et al., 2015; White & Ravelo, 2020), zonal SST gradient (Sadekov et al., 2013), Walker cell extent (Pausata et al., 2017), Inter-Tropical Convergence Zone (ITCZ) position (Pontes et al., 2022), among others (Z. Liu, 2002; Tudhope et al., 2001), a common mechanism explaining such ENSO behavior is hitherto missing.

The past 3.3 Ma encompass three key periods that have been under intense investigation due to their importance in improving our understanding of climate dynamics during warm states. First The mid-Pliocene warm period (referred as mid-Pliocene, ~3.2 Ma) featured a global mean temperature 2-4ºC higher than pre-industrial (PI) and elevated atmospheric CO2 concentrations (~400 ppmv) (Lunt et al., 2012). Over this period, proxy-data do not provide a clear picture on ENSO changes (Watanabe et al., 2011; White & Ravelo, 2020), but climate models systematically indicate reduced variability (Pontes et al., 2022). Other periods of interest are the Last Interglacial period (LIG, ~129 116 thousand years ago [ka]), the warmest

interglacial of the last 800 ka (Berger et al., 2016), and the mid-Holocene thermal maximum (MH ~8–4 ka). During both interglacials, atmospheric CO2 concentration was slightly lower than pre-industrial (PI) levels but with higher boreal summer insolation in the Northern Hemisphere. For both the LIG and the mid-Holocene, most proxy and models agree on reduced ENSO activity (J. Brown et al., 2020; Emile-Geay et al., 2016).

While paleo modelling studies of past warm climates suggest reduced ENSO variability, under a future high-emissions scenario models tend to simulate enhanced ENSO during the 21st century (Cai et al., 2018). Here, to investigate the effect of changes in Pacific mean state on ENSO variability, we analyze ENSO activity within the 4th phase of Paleoclimate and 6th phase of the Coupled Model Intercomparison Projects (PMIP4 and CMIP6, respectively).

## 2. Data and Methods

The PMIP4 project aims to gather the modelling community to consistently simulate key past periods in Earth's history. Here, we analyze 3 climate scenarios part of PMIP4: the mid-Pliocene (3.2 Ma, CO2=400ppm; Haywood et al., 2020), Last-interglacial (LIG, 127ka, CO2=275ppm, Otto-Bliesner et al., 2021), and mid-Holocene (6ka, CO2=285ppm, Brierley et al., 2020). In addition, we incorporate in our analysis projections for the 21$^{st}$ century under a high-emissions scenario (ssp585, CO2=400 to 1135 ppm,O'neill et al., 2016) and the scenario that abruptly increases atmospheric $CO_2$to four times pre-industrial levels (Supplementary Information Table S1). We use the pre-industrial control ($CO_2$=284 ppm; PI) simulations as reference for all scenarios.

To account for ENSO complexity, ENSO variability is decomposed into two types: C-index (Central Pacific; CP-ENSO), E-index (eastern Pacific; EP-ENSO; Takahashi et al., 2011), which have distinct anomalies centers in the equatorial Pacific and distinct impacts on remote areas. CP-ENSO variability is characterized by low intensity warming in the central-western Pacific, while EP-ENSO events are recognized by strong warming in the central-

eastern Pacific. As here we evaluate ENSO dynamics in climate models, the simulated anomaly centers of CP and EP-ENSO may vary across models. As such, C and E indices are obtained through combining the first two Empirical Orthogonal Functions of SST anomalies in tropical Pacific (15ºS-15ºN; 140ºE-80ºW). SST anomalies are computed by removing the monthly annual cycle. The first two principal components time series are combined to obtain the E ($(PC1-PC2)/\sqrt{2}$) and C ($(PC1+PC2)/\sqrt{2}$) indices (Supporting Information Fig. S1). SSTs of the ssp585 scenario are quadratically detrended before applying the EOF analysis. For completeness, we also evaluate SST variability in the Niño3 region (5ºS-5ºN; 150ºW-90ºW), which is projected approximately 45º between CP and EP-ENSO axes in the EOF space. Finally, the standard deviation of the time series of all indices are used as a proxy of ENSO amplitude (Taschetto et al., 2014).

Given that ENSO is a complex phenomenon of ocean-atmosphere interaction, not all climate models accurately simulate key processes of ENSO dynamics, resulting in poor ENSO simulations. To avoid being misleaded by results from models that do not capture ENSO dynamics, we apply two selection criteria based on the nonlinear Bjerknes feedback and nonlinear convective feedback, both of which contribute to the observed positive skewness of SST anomalies in the eastern Pacific (Appendix A and Supplementary Information Figure S2) (Cai et al., 2018; Dommenget et al., 2012). Applying these criteria results in 34 selected simulations of past climates and future scenarios out of the initial 87 simulations (excluding their respective PI simulations; Supporting information Tables S1 and S2).

## 3. Results

### 3.1 Mean state changes and ENSO activity

All three past warm climates (mid-Holocene, Last-interglacial and mid-Pliocene) share similar conditions in the tropical Pacific Ocean, though for different reasons (Brierley et al., 2020; Otto-Bliesner et al., 2021; Pontes et al., 2020). Firstly, they are characterized by intensified equatorial trade winds, especially in the central-western Pacific (Fig. 1a-c).

Intensified trades cause SSTs to be lower in the equatorial Pacific than in the surrounding area due to the Wind-Evaporation-SST (WES) feedback (Figure 1a-c) and strengthened equatorial upwelling. Secondly, the increased difference in inter-hemispheric warming leads to a northward shift of the ITCZ (Supporting Information Fig. S3). In contrast, the anthropogenic warming scenario (ssp585) shows an equatorially amplified warming with weakened trades and upwelling, conditions which are similar to that of El Niño events (Figure 1d). In these simulations, equatorially enhanced warming shifts the Pacific ITCZ and the South Pacific Convergence Zone (SPCZ) equatorward (Mamalakis et al., 2021;Supporting Information Fig. S3).

The response of ENSO in PMIP4 and CMIP6 simulations is likely tied to its mean state. Simulations of past warm periods show a near-unanimous weakening of ENSO variability and a more La Niña-like mean state, while the ssp585 simulations tend to show an increased ENSO variability with a more El Niño-like mean state (Figure 1e; all values computed in this study can be found in Supporting Information Table S1). In this context, studies have hypothesized that the strength of the coupled atmosphere-ocean climatological circulation can modulate ENSO variability (Pontes et al., 2022; Santoso et al., 2013), since, by definition, ENSO is a deviation from the mean climate.

### 3.2 Role of tropical convection centers

Important features that can modulate the strength of the coupled climatological circulation in the equatorial Pacific are the position of the atmospheric convection centers. There are two main convection centers in the tropical Pacific, the ITCZ and the SPCZ, whose meridional positions affect ENSO dynamics. Firstly, the more distant the convection centers are from the equator the reduced is ENSO growth and occurrence of extreme rainfall events (Pontes et al., 2022). Furthermore, the positions of the convection centers are key to determining the equatorial wind field and the associated ocean response. Convective regions are characterized by a convergent horizontal wind field, where horizontal wind velocities tend

to zero at their centers. Away from convective regions winds are more intense due to increased horizontal pressure gradients. Thus, the position of the convection centers determines the momentum transfer from the atmosphere to the ocean that could impact oceanic stratification and thermocline slope in the equatorial Pacific. These processes ultimately influence the effectiveness of the dynamical ocean-atmosphere coupling, important for ENSO development (Jin et al., 2006). For instance, a northward ITCZ displacement was found to be the main driver of reduced ENSO activity in the mid-Pliocene (Pontes et al., 2022), has modulated ENSO on a multidecadal timescale over the past 40 years (Hu & Fedorov, 2018) and is responsible for ENSO hysteresis under $CO_2$ removal scenarios (C. Liu et al., 2023). Finally, idealized experiments of changes in extratropical meridional SST gradients in the Pacific Ocean, which affect the ITCZ position, consistently impact ENSO activity (Chiang et al., 2008).

To evaluate the effect of the position of the convection centers on ENSO activity, the combined meridional displacement of the ITCZ and SPCZ is tracked during the developing and mature ENSO phases (austral spring-summer; Supporting Information Text T2). A preliminary result, which includes all models and simulations, indicates that the ENSO-convection centers relationship exhibits a quadratic shape (R2 = 0.35±0.01; Appendix B). This relationship indicates that there are two mean states that tend to inhibit ENSO development and there is an optimal distance at which the convection centers must be from the equator to maximize ENSO variability. Overall, poleward shifts and strong (>8º; combined absolute ITCZ and SPCZ meridional displacement) equatorward shifts of the convection centers are associated with reduced ENSO variability, whereas moderate (<8º) equatorward shifts increase ENSO activity. Applying the model selection criteria (see section 2) results in a further significant relationship that holds across different types of ENSO and metrics (Figure 2; further details and sensitivity tests are shown in Supporting Information Text T3 and Fig. S4).

The ENSO-convection centers relationship reveals important differences between CP- and EP-ENSO types. First, CP-ENSO activity is likely less sensitive to changes in the mean

state as indicated by a wider shape of the quadratic fit (a=-0.87, where a is the nonlinear coefficient of the quadratic model: $y(x) = ax^2 + bx + g$) and weaker relationship ($R^2 = 0.40$) compared to EP-ENSO (a = -2.18; $R^2 = 0.75$). Additionally, in our subset of models, CP-ENSO variability shows a non-robust increase under the ssp585 scenario (60% model agreement, Figure 2a). Nonetheless, if selecting models based only on the nonlinear Bjerknes feedback criteria, CP-ENSO show significantly increased variability (70% model agreement; Table S1 and S2), consistent with a previous study (Shin et al., 2022). The CP-ENSO sensitivity to different criteria may help explain inconsistent findings in previous CMIP phases (Cai et al., 2018; Taschetto et al., 2014).

On the other hand, EP-ENSO is strongly modulated by the position of the convection centers ($R^2 = 0.75$) and exhibits a high sensitivity to climate states (a = -2.18; Figure 2d). According to this finding, the climate system supports a maximum increase in EP-ENSO variance of approximately 47%, which is achieved with an overall equatorward displacement in the position of the convection centers of ~4.7º (Figure 2b). Finally, the Niño3 index likely captures a combination of CP and EP-ENSO variabilities ($R^2 = 0.65$, a = -1.47; Figure 2c) since its region encompasses both CP and EP anomalies (Takahashi et al., 2011).

### 3.3 3-D view of ENSO dynamics

In exploring the causes of the nonlinear ENSO sensitivity to background state, we find that both atmospheric and oceanic processes play an important role in explaining the ENSO-convection centers relationship. To investigate the effect of the position of the convection centers over convective anomalies, we first evaluate the frequency of extreme rainfall events (>5 $mm.day^{-1}$) in the eastern Pacific (Niño3 region) associated with the displacement of the convection centers (Figure 3a). We find that poleward shifts of the convection centers reduce the frequency of extreme events from one event per 9.4±3.6 years in the PI control to one event per 15.6±9.6 years, indicating suppressed convective feedback (Figure 3a). The convective feedback is further suppressed as the ITCZ shifts northwards until a full La Niña-like mean

state is reached and no extreme El Niño events occur (Pontes et al., 2022). Moderate equatorward shifts of the convection centers allow a more intense convective feedback and increase the frequency of extreme events to one event per 2.6±0.3 years (Figure 3a). The extreme scenario of strong equatorward displacement of the convection centers indicates a "permanent El Niño" situation, in which very intense (>9 mm.day$^{-1}$) rainfall events occur every year during austral summer (Figure 3a). In this simulation, continuous high rates of rainfall are associated with a small range (<2ºC) of SST variability (Figure 3a), indicating these events are not related to intensified convective feedback but are, in turn, reflecting the proximity of the climatological position of the convection centers to the equator. In fact, the climatological positions of the ITCZ and SPCZ in the most extreme simulation (3.4ºN and 3.5ºS, respectively) lie within the equatorial band (5ºS-5ºN), in agreement with an amplified warming in the eastern Pacific (dark blue plot in Figure 3a).

Another important process in ENSO dynamics is the easterly wind variability in the western Pacific (Niño4 region: 160ºE-150ºW). The intensity of the trade winds in the western Pacific has been shown to determine the amplitude of the wind variability in that region (Pontes et al., 2022), which is related to the intensity and frequency of westerly wind bursts, important for ENSO initiation (Chen et al., 2015). In this context, we found that the position of the convection centers also modulates the easterlies' variability through a quadratic relationship ($R^2$ = 0.52; Fig. 3b). The two scenarios that show reduced wind variability are consistent with reduced ENSO activity. Poleward movement of the convection centers increases the horizontal scale of the wind flow that reaches the western Pacific, generating wind anomalies in the South Pacific Subtropical High region, indicating that this new regime is more geostrophic, therefore more linearly balanced and deterministic (Fig. 1a-c). On the other hand, strong equatorward shift of the convection centers requires weak horizontal wind intensity, consistent with weak wind variability (Fig. 3b).

Finally, to investigate the possible modulation of the position of the convection in the dynamical coupling between the ocean and the atmosphere, we analyze the response of the wind-thermocline coefficient (Jin et al., 2006). This parameter measures the sensitivity of the tilt mode of thermocline slope anomalies to wind stress anomalies, which during El Niño events result in eastward heat advection by downwelling equatorial Kelvin waves (Timmermann et al., 2018). Our results indicate that the displacement of the convection centers also modulates the wind-thermocline coefficient through a quadratic relationship ($R^2 = 0.48$; Figure 3c). Strong equatorward shifts of the convection centers reduce the effectiveness of wind anomalies in generating swings of the thermocline. In such scenarios, the propagation of Kelvin waves in a flatter thermocline does not effectively promote oscillations between El Niño and La Ninã states. Poleward shifts of the convection centers increase the climatological thermocline slope, requiring stronger wind to anomalies to promote thermocline oscillation, and thus also reducing the dynamical air-sea coupling.

## 4. Conclusion

The results described above suggest a 3-dimensional view of ENSO complexity by linking meridional shifts of the convection centers, atmospheric convection, ocean stratification, and zonal thermocline oscillations. The interplay of these processes results in a nonlinear ENSO sensitivity to mean states, encompassing three key background conditions (Figure 4). A poleward migration of the convection centers intensifies the easterlies in the equatorial Pacific, weakening the convective feedback. Increased momentum transfer to the upper-ocean reduces ocean stratification, ultimately resulting in a weaker dynamical coupling, hampering ENSO development (Figure 4). These background conditions resemble a La Niña-like mean state. Moderate equatorward shift of the convection centers (<9º) reduces the intensity of the equatorial trades, increasing upper-ocean stratification, and consequently amplifying the dynamical coupling, which allows the equatorial trades and thermocline slope to be rapidly reversed, enhancing ENSO activity and extreme rainfall events (Figure 4). Finally,

a strong equatorward shift of the convection centers creates a permanent El Niño-like mean state in the eastern Pacific. The fact that the convection centers lie at the equator do not allow momentum transfer to the ocean, resulting in a highly stratified ocean and dampened dynamical coupling. Nonetheless, this scenario is associated with intense warming in the eastern Pacific that enhances the climatological thermodynamical coupling, where climatological high SSTs allow intense rainfall every year.

It is important noting that these results are subject to systematic climate model biases. The main common biases that could affect the processes here analyzed are biases in tropical convection and SST, such as double-ITCZs, overly strong cold tongue and predominantly zonally oriented SPCZs (Narsey et al., 2022; Tian & Dong, 2020). Nonetheless, to reduce the influence of such biases, we applied two model selection criteria. Additionally, the artificial calendar effect, due to modified orbital parameters, can reduce the accuracy for LIG and mid-Holocene simulations (Bartlein & Shafer, 2019).

Climate models suggest that the projected increase in ENSO variability has likely not occurred in the past ~3.3 Ma. While there are significant uncertainties in both paleoclimate simulations and paleo-proxy records, both tend to suggest reduced ENSO variability during the LGM, the LIG and the mid- to late Holocene (Brown et al., 2020; Emile-Geay et al., 2016; Sachs et al., 2009). While these results agree with our understanding of past ITCZ changes (Sachs et al., 2018; Schneider et al., 2014), past dynamics of the SPCZ need to be better understood and constrained (Brown et al., 2020). Nonetheless, state-of-the-art climate models and proxy data initially suggest that the combination of global warming and increased extreme weather events is likely unique to the 21st century. This may reduce the resilience of many species and their adaptation to these unprecedented climate conditions, alerting the global community to a possible great biodiversity loss.

**Acknowledgments**

For their roles in producing, coordinating, and making available CMIP6 and PMIP4 model output, we acknowledge the climate modeling groups (Supplementary Information Table S1), the World Climate Research Programme's Working Group on Coupled Modelling and the Global Organization for Earth System Science Portals. GMP and PLDS acknowledge funding from the São Paulo Research Foundation (grant number 2021/11035-6). This work is supported by the Australian Research Council Special Research Initiative, Australian Centre for Excellence in Antarctic Science (project number SR200100008).

**Data and code availability**

Simulation from pre-industrial control, high emissions scenario (ssp585), mid-Holocene, Last Interglacial, and CESM2, EC-Earth3-LR, NorESM1-F, IPSLCM6A and GISS-E2-1-G simulations of the mid-Pliocene can be obtained directly through the Earth System Grid Federation repository (ESGF; https://esgf-node.llnl.gov/search/cmip6/). Other mid-Pliocene simulations are available upon request to Alan M. Haywood (a.m.haywood@leeds.ac.uk). Models used in each analysis were selected based on data availability in their respective databases. The last 100 years of each model's simulation are used. All climate periods (paleoclimates and projections) are compared to the pre-industrial climate. Computer codes are available upon request to Gabriel M. Pontes (gabrielpontes@usp.br).

**Author contributions**

GMP designed the study, conducted the analysis, prepared all figures, and wrote the original manuscript. PLDS and LM contributed with discussions and commented on the manuscript.

**Competing interests**

At least one of the (co-)authors is a member of the editorial board of Climate of the Past.

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

**Figures**

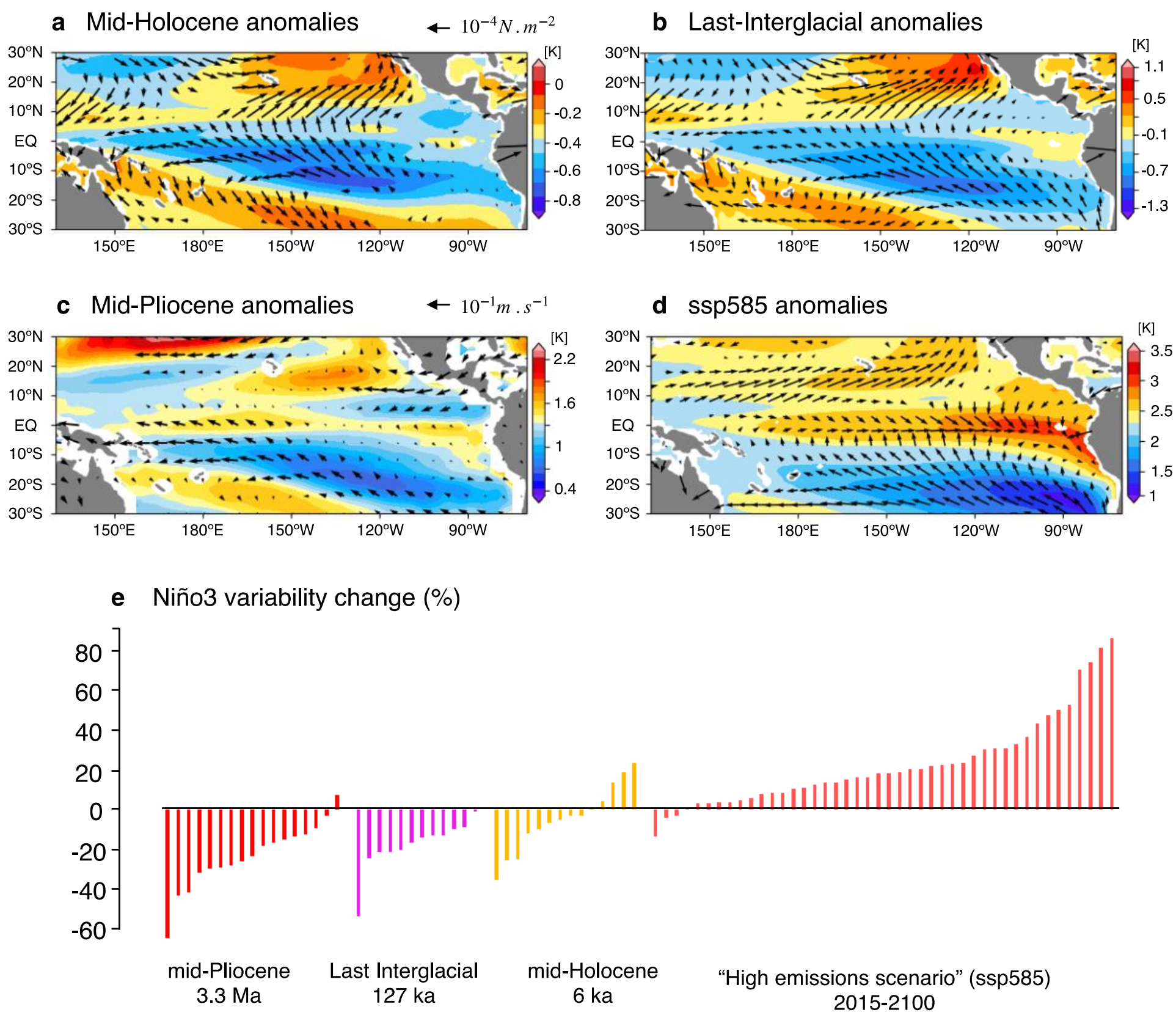

**Fig. 1. Paleoclimate anomalies and ENSO variability change. a-d** Multi-model annual mean change in SST and wind field for the mid-Holocene (**a**), Last-interglacial (**b**), mid-Pliocene (**c**) and high emissions scenario (ssp585; **d**). The SST colorbar varies for each panel. The colorbar is set to indicate regions in which SST changes are greater (warm colors) and lower (cold colors) than the mean tropical Pacific warming in each set of simulations, making changes in SST gradients easily identified. Arrows represent wind stress changes in panel **a**, **b** and **d** and, due to data availability, surface (10-meter) winds in panel **c**. Arrows are plotted where there is a significant change in either zonal or meridional component. That is, at least a 70% model agreement in the sign of the change. **e** Change in niño3 standard deviation for each simulation used in this study. The pre-industrial control simulation is used as reference when quantifying changes. The value for each simulation can be found in Supplementary Information Tables S1 and S2. Maps in this figure were plotted using the cartopy Python library.

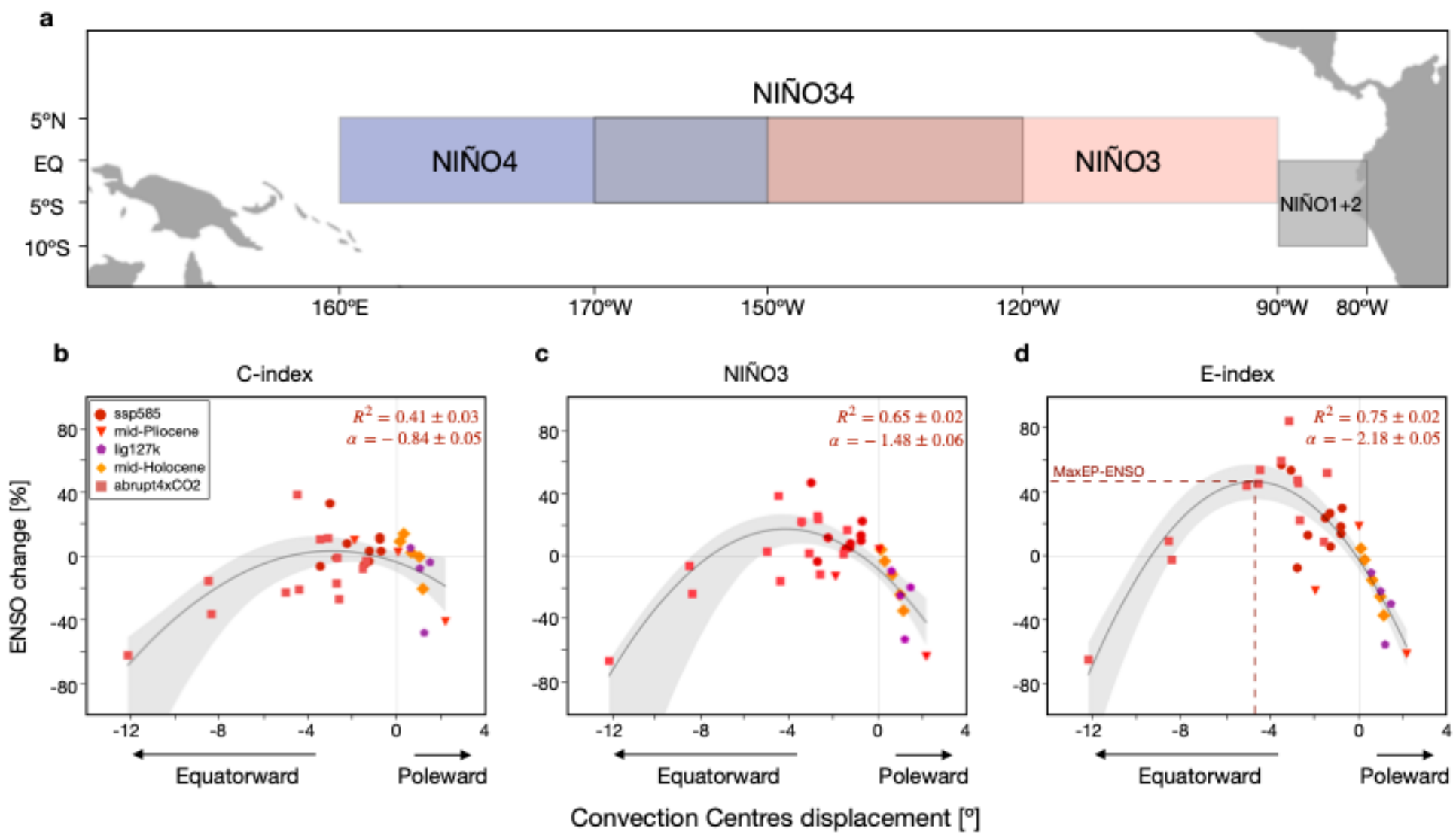

**Fig. 2. ENSO-Convection Centers relationship.** **a** Location of key ENSO regions. **b-d** Dispersion diagram between the overall displacement in the meridional position of the ITCZ and SPCZ and ENSO indices. **b** C-index. **c** E-index. **d** Niño3 index. The solid black line indicates the quadratic fit based on the least squares method. Banding indicates 95% confidence interval based on a 1000-sample bootstrap. The mean displacement of the convection centers boreal spring-summer is considered (i.e., encompassing developing and mature ENSO phases). $R^2$ indicates the coefficient of determination and a the nonlinear coefficient of the quadratic regression model. Error estimates for $R^2$ and alpha we calculated as one standard deviation of 1000 bootstrap realizations. In panel **d** the dashed line indicates the maximum ENSO variability supported by the climate system (MaxENSO). Maps in this figure were plotted using the cartopy Python library.

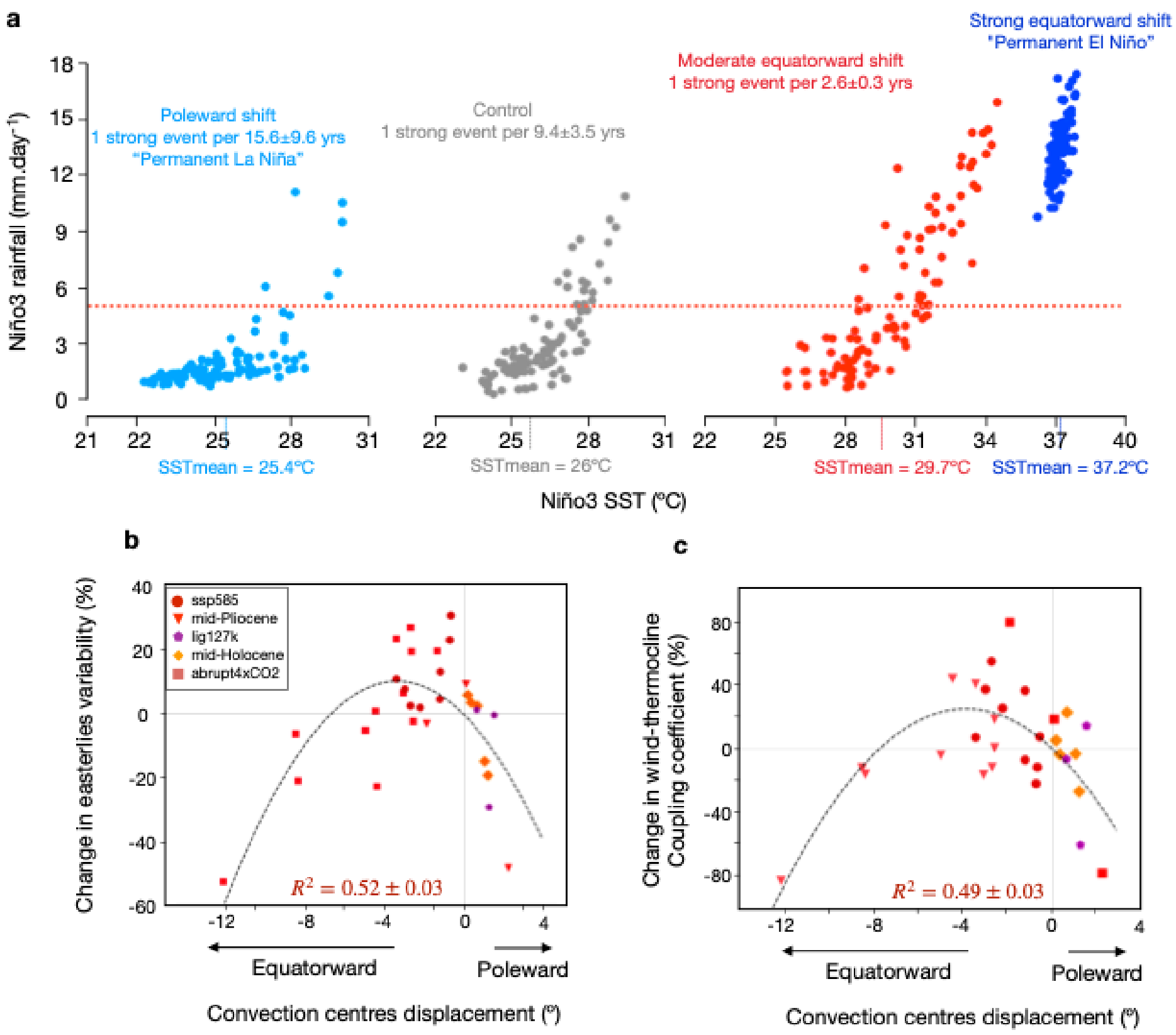

**Fig. 3. Equatorial Pacific climate and ENSO feedbacks. a** Nonlinear convective feedback: relationship between DJF Niño3 SSTs and DJF Niño3 rainfall. Models were grouped into 3 subgroups according to the magnitude of the shift of the convection centers: poleward (>0º), and moderate (<9º) or strong (>9º) equatorward displacements. The 1000 samples of 100 years were obtained through the bootstrap method (Methods). The plot shows the first 100-years sample of each group. The frequency of extreme events is indicated by the mean frequency of events that exceeded 5 mm.day$^{-1}$ across all realizations (orange dashed line). Error estimates are indicated by the standard deviation of the realizations. **b** inter-model relationship between the displacement of the convection centers and change in easterly winds variability. **c** inter-model relationship between the displacement of the convection centers and the wind-thermocline coupling coefficient. $R^2$ indicates the coefficient of determination of the quadratic regression model. Error estimates are given by the standard deviation of 1000 bootstrap realizations.

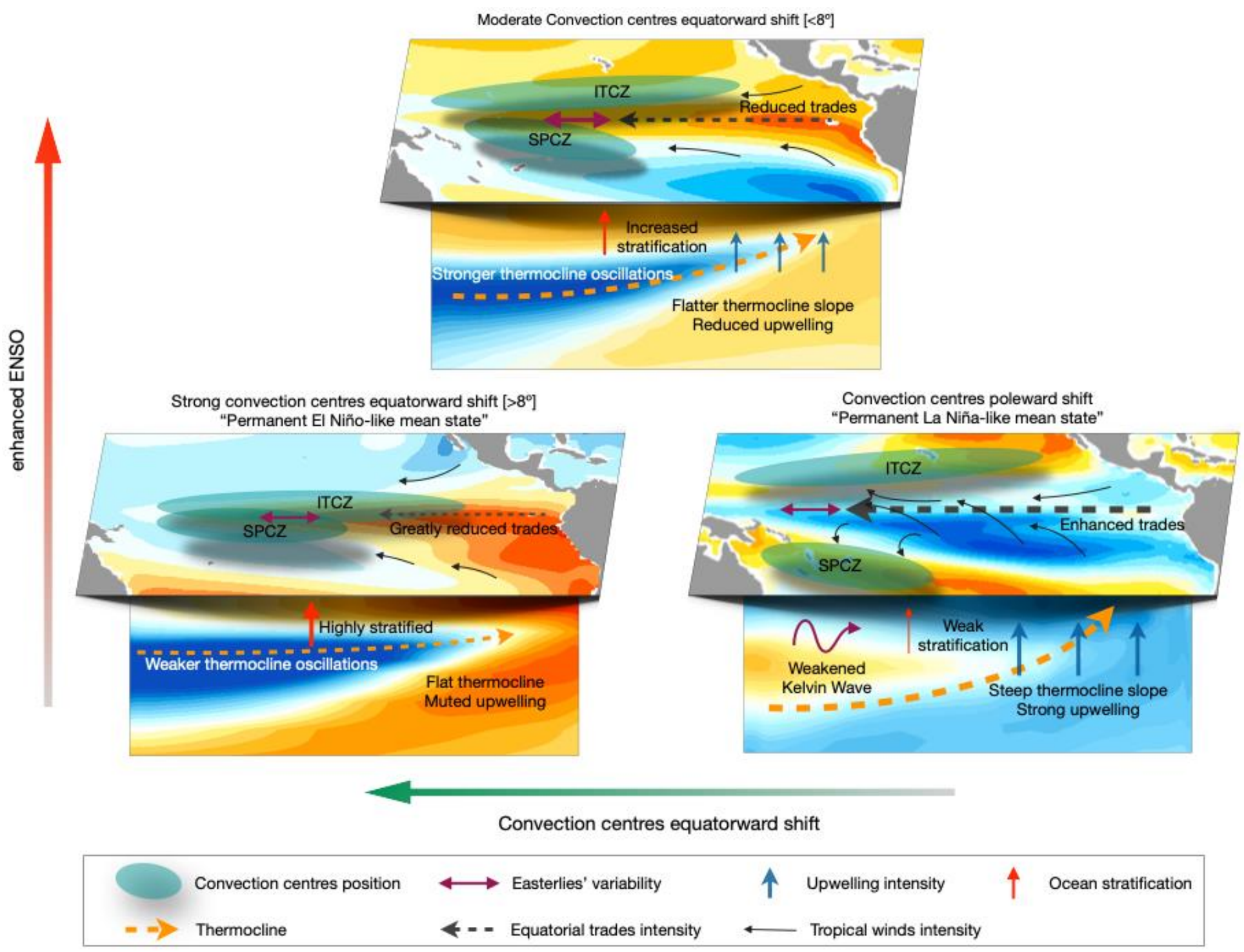

**Figure 4 | Mechanisms for changes in ENSO activity across climate states.** Schematic of mean state features associated with ENSO variability. Bottom right: A poleward shift of convection centers enhances equatorial coupled circulation and reduces the dynamical coupling and ENSO activity. Bottom left: A full equatorward shift of the convection centers results in a very weak coupled circulation, flat thermocline, a highly stratified upper-ocean and weak dynamical coupling, thus reducing ENSO activity. Top: A small equatorward shift of the convection centers decreases equatorial coupled circulation and increases kelvin wave energy, increasing thermocline swings and ENSO variability. All comparison statements are related to the pre-industrial period. Maps in this figure were plotted using the cartopy Python library.

## Appendix A

The model's ability to simulate the nonlinear Bjerknes feedback is assessed through the nonlinear relationship between the first two principal components of monthly SST anomalies in the tropical Pacific. Models were required to simulate the parameter a, given by the nonlinear coefficient of the fitted quadratic model, greater than half of the observed value ($a_{obs}$=0.32, Supporting Information Fig. S3) (Takahashi et al., 2011). The criteria used to select models that correctly capture the strength of the convective feedback is based on an essential definition

of extreme ENSO-related rainfall events, which are defined as precipitation events greater than 5 mm/day$^{-1}$ in the Niño3 region. In observations, the 5 mm/day$^{-1}$ rainfall rate is achieved at an SST anomaly of 2ºC, which gives a convective feedback of 2.5 mm/day$^{-1}$/ºC$^{-1}$ (Supporting Information Fig. S2). To ensure that models capture the observed strength, models were required to simulate convective feedback greater than 2 mm.day$^{-1}$.ºC$^{-1}$ in their pre-industrial runs. The model's ability to properly simulate ENSO skewness filters out models that systematically simulate overly wet (i.e., double-ITCZs) and dry conditions (i.e., overly strong cold tongue) in the eastern equatorial Pacific. These models simulate SSTs well below or above the convective threshold of 26-28ºC (Johnson & Xie, 2010), thus simulating unrealistic convective feedback.

The maximum precipitation regions are used as a proxy of the meridional positions of the ITCZ and SPCZ. The ITCZ position is taken as the average latitudes over which precipitation in the tropical North Pacific Ocean (0º-20ºN) is greater than 50% of the maximum zonally averaged precipitation over 120ºE-90ºW. The position of the SPCZ is obtained in a similar way but considering the tropical South Pacific (20ºS-0º). This methodology captures migrations of ITCZ and the SPCZ independently from one another. Given that our objective is to quantify their overall displacement with respect to the equator, this is calculated as their absolute shift:

$$D = [\,|ITCZ_S| - |ITCZ_{PI}|\,] + [\,|SPCZ_S| - |SPCZ_{PI}|\,]$$

where the subscript 'S' denotes paleo (mid-Pliocene, LIG, LGM or mid-Holocene) or projection (ssp585) scenarios, while the subscript 'PI' denotes the pre-industrial simulation used as reference. A negative (positive) displacement (D) indicates an overall equatorward (poleward) shift. It is important noting that double-ITCZ biases may affect the SPCZ position. The double-ITCZ bias is an artificial feature produced by most climate models that overestimates the tropical precipitation south of the equator in the central-eastern Pacific.

The efficiency of the dynamical coupling between the ocean and the atmosphere is measured through the intensity of the wind-thermocline coupling coefficient (Jin et al., 2006). This coefficient measures the sensitivity of the tilt mode of thermocline slope anomalies to wind stress anomalies, which during El Niño events results in eastward temperature advection by downwelling equatorial Kelvin waves:

$$\langle h \rangle_E - \langle h \rangle_W = \beta_h \langle \tau_x \rangle$$

where *h* indicates the thermocline depth, $b_h$ the wind-thermocline coupling coefficient and $t_x$ the zonal wind stress. Subscripts 'E' and 'W' denotes area average in the eastern (5ºS-5ºN; 150ºW-90ºW) and western (5ºS-5ºN; 160ºE-150ºW) equatorial Pacific, respectively. The thermocline depth is computed from the mean temperature profile in each of the boxes indicated above. This is the weighted average depth, based on depths in which the temperature gradients are greater than 50% of its maximum (Pontes et al., 2022). The wind-thermocline coefficient is computed from monthly anomalies, which capture the evolution of the thermocline slope within each single ENSO event.

**Appendix B**

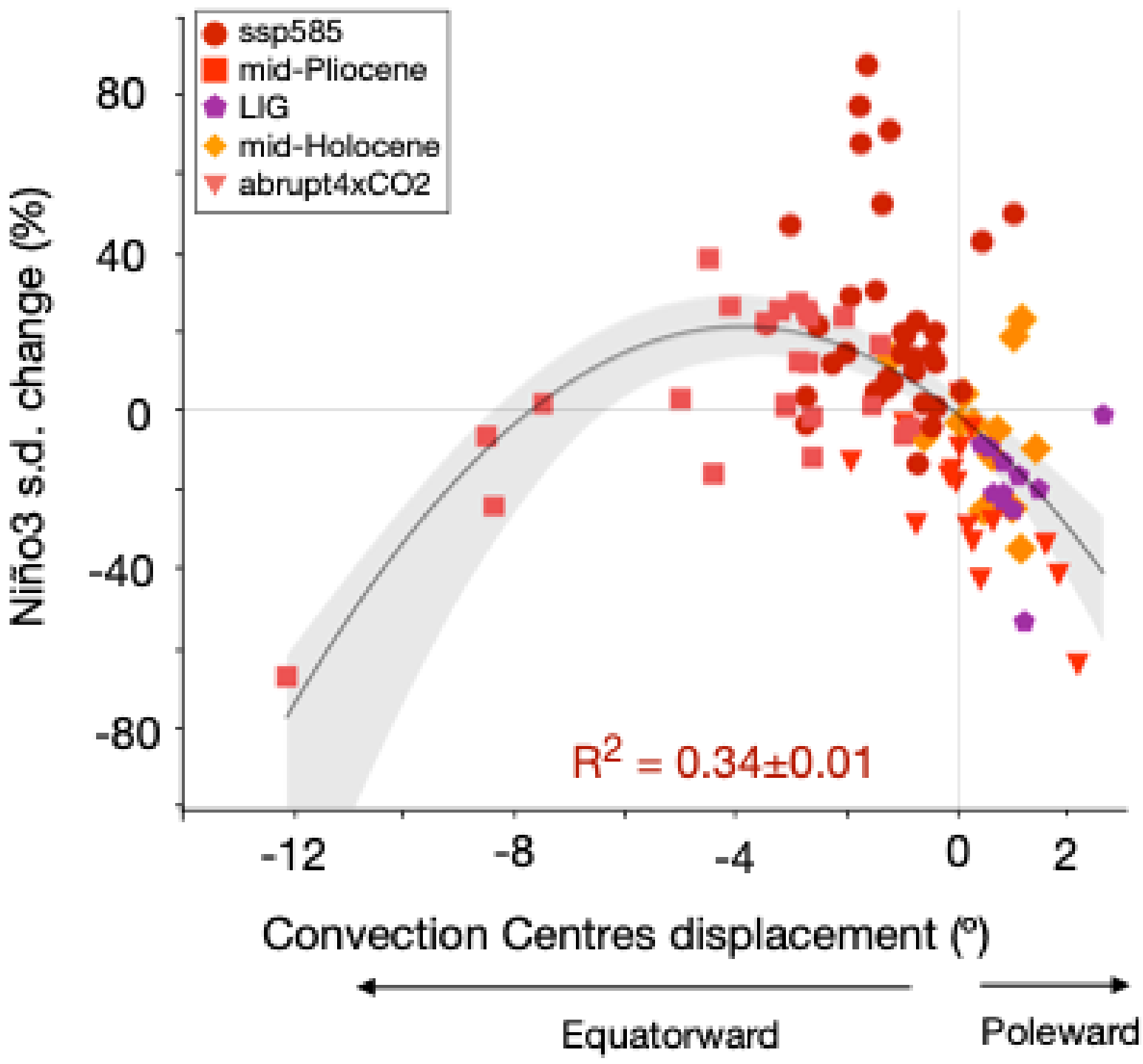

**Appendix B Fig. 1B. ENSO-convection centers relationship.** Dispersion diagram between the overall displacement in the meridional position of the ITCZ and SPCZ and the Niño3 index. The solid black line indicates the best quadratic fit based on the least squares method. Banding indicates 95% confidence interval based on a 1000-sample bootstrap. The mean displacement of the convection centers boreal spring-summer is considered (i.e., encompassing developing and mature ENSO phases). $R^2$ indicates the coefficient of determination and a the nonlinear coefficient of the quadratic regression model. Error estimates for $R^2$ and alpha we calculated as one standard deviation of 1000 bootstrap realizations. No model selection criteria were applied.