# Peer review of "Rapid communication: Nonlinear sensitivity of El Niño-Southern Oscillation across climate states"

_EGUsphere, 2024_

## Referee Comment (RC2)

Review of:
**Nonlinear Sensitivity of El Niño-Southern Oscillation across Climate States**
By Pontes et al.
Submitted to Climate of the Past

**Summary**
In this manuscript, the authors aim to clarify the relationship between properties of ENSO variability and the mean background climate in a range of different past and future warmer climate states. The authors employ different past climate simulations (covering the mid-Pliocene, last interglacial and mid-Holocene) and future simulations (ssp585 and abrupt 4x CO2) and present their results with respect to pre-industrial climate simulations. The results show a common mechanism linking changes to the mean atmosphere-ocean circulation to an increase or decrease in ENSO variability. In fact, the authors reveal a non-linear relationship between a change in ENSO variability and the change in the meridional positions of the intertropical and South Pacific convergence zones. Three distinct regimes of ENSO variability change are identified and the processes in the tropical Pacific mean climate leading to the existence of these regimes are explained.

I believe this is a very interesting study with relevant findings. The scientific content is substantial and provides a significant contribution to the field of ENSO complexity. The analysis is sound, and the figures are very clear. I do have a few issues, related to the data selection and to the Introduction, that I would consider major (but should not require too much effort to resolve). Apart from those, I present minor comments and a few technical corrections. I recommend this manuscript to be published in Climate of the Past subject to revisions.

**Major concerns**
My main issue is the following. Your aim is to analyze nonlinear ENSO sensitivity to the mean climate. I have no issues with that with regards to the past climate simulations. But the future simulations, ssp585 more so than abrupt4xCO2, are transient simulations, and not in equilibrium. You seem to use data from 2015 - 2100 (if I am correct), but both the mean climate as well as ENSO characteristics change quite drastically over that period. I don't think it makes sense to take the mean climate over this whole period and to consider that ENSO characteristics are constant over this period, because they are not. Either you need to convince me that what you are doing is fine (e.g. show me that the mean climate and ENSO properties between f.e. 2015-2060 and 2061-2100 are similar enough for your analysis following some statistical test), or I would suggest you use different simulations or a different selection of years. My suggestion would be to use the simulations from the LongRunMIP project instead of the ssp585. LongRunMIP also encompasses the abrupt4xCO2 simulation, so that works out well. It includes high CO2 'future' simulations, but equilibrated and not transient, so the mean climate can be considered in a mean state, and ENSO can be considered not changing over the last 100 years. Another option is to use extended SSP585 runs, but I don't think many models ran those runs, so that might give you only a small ensemble to use.

I have an issue regarding reproducibility of your results because is it quite unclear what your specific data selection is. Which observational dataset is used? It is not mentioned in the main text, nor in the appendix, nor in the supplement. Do you use monthly data? And how many years? I find 100 years in "Data availability". Are the mean climate features you show based on the mean of those 100 years? For the past periods that is fine. But most ssp585 simulations go from 2015 to 2100; does that mean you only use 85 years? The caption in Figure 1e seems to suggest so. What about the abrupt 4xCO2 simulations though? I am left to find out a lot of details about your data selection from different parts of the paper. I find that a major issue, but I think it will cost you very little time to collect those details in the Methods section.

Lastly, I have some issues with the Introduction. It is clear, but incomplete (in my view). I have three recommendations:

a) First, I find it unclear from the Introduction what the research aims are. Currently, the only stated aim is in L66-67: "to investigate the effect of changes in Pacific mean state on ENSO variability." That is quite vague. Can you explain in more detail: What the main research gap is that you plan to address / what the main research aims are, what your research questions are (and why – relate this to the literature overview), and how you plan to answer those (i.e. more detail on the specific features you will analyze).

b) Too much important background information is explained after the Introduction (i.e. in the Methods and Results). For example, the different types of ENSO are only explained in the Methods – why not in the Introduction? The SPCZ is mentioned first in the Results (L118), but plays an important role throughout the paper – why it is not explained in the Introduction? The most striking is the first paragraph in section 3.2 (L129 – 147). This paragraph is a literature overview, and harbors no results, so should not be included in the Results. Please incorporate it in the Introduction.

c) The literature overview on ENSO variability in the future is incomplete. Cai et al. (2021) present a literature review on ENSO in a warm climate and show (with more confidence than the study you cite now) that ENSO amplitude increases under future warming, but also that equatorial Pacific rainfall intensifies, extreme ENSO rainfall events increase, and EP ENSO variance increases clearly while agreement on CP ENSO variance increase is low. I believe all these results resonate with your findings, so it is relevant to mention in the Introduction. Additionally, while near-term future climate simulations (e.g. until 2100) show an intensification of ENSO variability, equilibrated long-term warming simulations show a robust decrease (Callahan et al. 2021 and Zheng et al. 2022). Both studies employ the LongRunMIP ensemble simulations, that represent equilibrated 2x, 4x, 6x and 8x CO2 increases, both abrupt and 1pct increase type simulations. Almost all these simulations show a robust decrease in ENSO variability. I find that a very significant finding to mention with regards to your study. Additionally (as mentioned in my previous point), I would suggest including these results as they provide suppressed ENSO not in the past but in a long term future, and I would be very interested to see in which of the nonlinear regimes the LongRunMIP results end up.

Cai, W., Santoso, A., Collins, M. *et al*. Changing El Niño–Southern Oscillation in a warming climate. *Nat Rev Earth Environ* **2**, 628–644 (2021). https://doi.org/10.1038/s43017-021-00199-z

Callahan, C.W., Chen, C., Rugenstein, M. *et al*. Robust decrease in El Niño/Southern Oscillation amplitude under long-term warming. *Nat. Clim. Chang.* **11**, 752–757 (2021). https://doi.org/10.1038/s41558-021-01099-2

Zheng, Y., Rugenstein, M., Pieper, P., Beobide-Arsuaga, G., and Baehr, J.: El Niño–Southern Oscillation (ENSO) predictability in equilibrated warmer climates, Earth Syst. Dynam., 13, 1611–1623, https://doi.org/10.5194/esd-13-1611-2022 , 2022

**Minor comments**

I feel that L39-40 "which has been … instrumental period" feels out of place here. I suggest removing it here (just end the sentence) and combine it with "with strong impacts worldwide" in L36. So, end L36 at "eastern Pacific (ref)." then insert: "It has been the most important driver of year-to-year climate variability during the instrumental period, with strong impacts worldwide."

Subscripts and superscripts need to be included consistently. CO2 should be $CO_2$ in L60, L73, L74, L76 and throughout. You write 21st, 4th and 6th in L66-68, but 21$^{st}$ in L75. Choose a style and apply consistently. In L152 R2 should be $R^2$.

L60 "both interglacials" – I understand you mean LIG and MH. However, the mid-Pliocene simulations you use from PlioMIP2 are also an interglacial period. I feel that is a detail worth mentioning (maybe in the Methods where you introduce PlioMIP2, explain that the simulation protocol is tuned to the KM5c interglacial).

L80-82: I think you need to rewrite this sentence. C-index is an index that represents the CP-ENSO variability, but now it reads as if C-index is the name of a type of ENSO variability (which is, in fact, CP ENSO).

L91: only the ssp585 data is detrended? I would say, for consistency, to apply detrending to all simulations (although it will likely not affect the results substantially).

L103-105 I would like to see included in the text how many models you end up using per scenario/past period after the selection. From the supplement it seems that you end up with quite some models for the future simulations, but only a relatively small selection for the past periods. I find that relevant information to include here.

L112 WES feedback could merit a citation

L113 "the increased difference in inter-hemispheric warming" -- this is not something that is necessarily shown by your results, is it? the SSTs in a-c mainly show a reduced meridional gradient. What I think you mean is the heat asymmetry between both hemispheres, correct? I know from your previous work on the Pliocene (Pontes et al 2022) that this heat asymmetry, stemming from planetary albedo differences related to

vegetation, ice sheets, or orbital differences, drives the northward shift of the ITCZ. For the LIG and MH, I am left to deduce the cause of this inter-hemispheric difference from a sentence in the Introduction (L60-62): "higher boreal summer insolation", which causes a difference in interhemispheric warming. This is a lot of deduction work for a reader, so I think this merits a sentence here explaining in more detail what is meant by 'increased difference in interhemispheric warming' and clarify this is based on previous findings and not on results in this work.

L118 regarding 'equatorward' and relating to the other reviewer's comments: to me it is quite clear that the ITCZ in the Pacific is always located in the Northern Hemisphere, so I understand your usage of equatorward and poleward here. However, that might not be obvious to all reader, so maybe you can clarify this point.

L124-126 "In this context … mean climate" -- I am not convinced by this formulation. I think there are two different formulations in this sentence, one more related to physics (which I suggest keeping), and the latter more related to how ENSO is defined. Since ENSO is defined as an anomaly to a mean climate, it indeed would follow that when a mean climate is different, ENSO will be different. But that to me feels quite arbitrary and does not reveal any new information on the physical link between the mean state and ENSO variability, which is what you are trying to establish in this study. The first part is more interesting: the strength of the mean atmosphere-ocean circulation can modulate ENSO variability. If possible, I would expand more on that, e.g. a strong mean circulation (La Niña like mean state) inhibits the development of anomalies, effectively suppressing ENSO variability, while a weak mean circulation (El Niño like mean state) is more favourable for the development of anomalies that can lead to ENSO events.

L131-132 "whose meridional positions affect ENSO dynamics" – I think this statement merits a reference

L151-152 "the ENSO-convection centers relationship" – propose to change this to "the relationship between ENSO amplitude change and the displacement of the convection centers"

L160 (regarding the previous paragraph): I agree with the other reviewer that it seems like the abrupt4xCO2 results are important for establishing a quadratic relationship over a linear one. However, I think you already showcase that even without the abrupt 4xCO2 simulations, a quadratic fit is better than a linear fit for all ENSO indices (namely in supplement figure S4). It would be worth it to mention this result in the main text.

L161-L165: Are these findings based on the results from all models or only the selected subset? Do these results relate to the curves shown in Figure 2? Please clarify

L163-165: the description of the quadratic model could be moved to the Methods

L165-L169: you use 10 models for the ssp585 scenarios. is the difference between 6/10 agreeing or 7/10 agreeing what you call 'non robust increase' and 'significantly increased' ..? also, I am confused where you state 'in our subset of models' and in the

next sentence 'if selecting models based on …'. Isn't this subset you mean already only those models that pass this criteria test?

L195 "SST variability" I think you mean that there is a small model spread in the mean SST in the Niño3 region. A small range in SST variability would suggest that most models agree on the amplitude of ENSO, which is not what you seem to imply.

L213 (regarding previous paragraph): following these results, I would be interested to see the ENSO amplitude change as a function of the change in easterlies variability. It would probably show a linear relationship where a weaker (stronger) ENSO relates to a weaker (stronger) easterlies variability. However, it would not show that the weaker ENSO / weaker easterlies variability is two different regimes: where the convection centers are shifted poleward or strongly equatorwards. Your results and conclusions could harbour a recommendation to the ENSO research community to not just look at this ENSO change - easterlies variability relationship, but also investigate the shift in convection centers, if one truly wants to know which regime in ENSO suppression is occurring. I think that is a valuable recommendation to state here on in the Conclusions).

L231 "three key background conditions": but these are background states with regards to a PI control, so I would suggest calling it something like 'anomalous background states', or 'three distinct regimes deviating from the PI control'.

L231 related to this, you follow up with mentioning these three regimes. For clarity for the reader, I would suggest numbering these regimes, e.g. in L232 1) A poleward migration …, L236 2) A moderate equatorward shift… and L239 3) A strong equatorward shift (remove "finally")

L238: "…dynamical coupling." – I would add ", and thus dampened ENSO variability." To that sentence for clarification.

L249-250: "Nonetheless, … selection criteria." – I find this a strange read, and it should/could include a suggestion for the community. Propose to rewrite: "In this study, we applied two model selection criteria to reduce the influence of such biases, and we recommend others to employ similar model selection steps when investigation ENSO complexity."

L261-263: "This may … biodiversity loss." – I find this a highly suggestive sentence which seems to be included to increase the relevance of this study. I would suggest removing it. The study is relevant in its own right. I would recommend including a different take-away (more related to the study results), or just remove this sentence altogether.

Figures:

Figure 1: Am I correct in assuming panel e shows all model simulations, and not just the models selected based on the criteria mentioned in the methods? I think it makes sense to include all models here, since you are just looking at mean climate and ENSO and not

yet the relationship between the two. However, I would clarify in the Methods that you include all models for this result, and only move forward with a smaller subset for the later results.

Figure 1: you might want to extend the arrows with the correct units to all panels (a-d) so that it is clear from the figure itself which panels show wind stress, and which show surface wind. Now, from the panels itself it is not clear (although the caption clarifies).

Figure 2a: Is it really necessary to include this figure with the Niño regions..? Consider leaving it out.

Figure 3a: am I correct in assuming that 'control' here means PI control data, and all the other data is the grouped results of all the past and future simulations? if yes, are all model simulations included, or only the subset of models that pass the criteria test..? Please clarify

Figure 3a: 'strong equatorward' is defined as >9deg. looking at fig 3b, to me that seems like only 1 simulation belonging to abrupt4xCO2. Is that correct? Because that would imply that your findings on the strong equatorward rainfall-SST relationship is based on 1 model and 1 simulation, making it hard - I think - to connect meaningful conclusions to that result.

Figure 3c: if I am correct, the triangles and squares in panel c are the wrong way around. Please change accordingly

Figure 4: I would suggest naming these panels (a), (b) and (c).

**Technical corrections**

L29: El Niños → El Niño events
L31: how tropical → how the tropical
L41: likely the → likely that the
L61-62 "in the Northern Hemisphere" can be removed ("boreal" already implies this)
L66: in Pacific → in the Pacific
L130: position → positions
L141: coupling, important → coupling, which is important
L205: In this context, we found → We find
L228: The results described above → Our results
L236: Moderate → A moderate
L274: Simulation → Simulations
L281: the pre-industrial climate → pre-industrial climate simulations

---

## Author Response (AR2)

Dear Reviewers,

We would like to thank you for the useful comments and suggestions that permitted us to improve our manuscript.

Our response to each comment is given in a point-by-point manner below, with the Reviewers' comments in black, our responses in blue and excerpts of the main text where each comment has been addressed in purple. We list below the main changes we made in the manuscript.

1. The 'Introduction' was modified to clearly spell out the aims of our study, include the necessary background information and identify the literature gaps.

2. We now include only scenarios that would represent a near-equilibrium state. For this end, we have removed the evaluation of the 21$^{st}$ century SSP5-8.5 scenario and now evaluate this scenario from years 2251 to 2300. We have also incorporated the SSP1-2.6 scenario considering the same time slice. Figures 1, 2 and 3 have been updated accordingly.

3. Nevertheless, we present an evaluation of the SSP3-7.0 scenario from years 2051 to 2100 in the Appendix to illustrate the proposed mechanism under a transient scenario.

4. We have now included the description of convection-centers index in main text.

5. Analyses of the central and eastern ENSO types has been moved to Appendix.

Yours sincerely,

Gabriel M. Pontes, on behalf of all authors

**Reviewer 1**

Pontes and coauthors present a compilation of climate model experiment results from past and future warm states (the mid-Holocene, the Last Interglacial, the mid-Pliocene, SSP5-8.5, and $4xCO_2$) to investigate the mechanisms that drive El Niño-Southern Oscillation (ENSO) variability across changing background climate conditions. Taken together, the authors suggest the models support a common underlying mechanism influencing ENSO behavior: the latitudinal position of the Intertropical and South Pacific Convergence Zones (ITCZ and SPCZ, respectively). This response is tied to changes in the surface wind field and its associated impacts on the upper ocean, which can either enhance or weaken the ocean-atmosphere coupling strength that underpins ENSO development. Importantly, the authors find that this response to ITCZ and SPCZ position is non-linear. Peak ENSO suppression occurs when both convection centers are shifted equatorward by ~4 – 5° latitude. Further equatorward migrations, such as in the $4xCO_2$ experiment, lead to permanent "El Niño-like" conditions that weaken ocean-atmosphere coupling and ENSO intensity. Likewise, poleward displacements by >8° latitude weaken the convective feedback, which also reduces ENSO intensity. The authors also find that this response is stronger for east Pacific ENSO than central Pacific ENSO. Altogether, the modeling results reveal negative quadratic relationships between convective center displacement and ENSO behavior, variability in easterly wind bursts, and wind-thermocline coupling, supporting a non-linear dynamic between ENSO and background climate.

Overall, I think the analysis presented in this paper is interesting and certainly a worthy contribution to the literature. I do have some major comments regarding the background climate conditions represented in the compilation and whether the authors' proposed mechanism finds support in the paleo record. Otherwise, most of my suggestions are minor/editorial and could be triaged by the authors with ease.

We thank the Reviewer for their constructive comments, which helped improve our manuscript. Below we answer the reviewer's comments and include in purple some excerpts of modified text to be included in the revised manuscript.

**Major Remarks**

1.      What immediately stood out to me was that the modeling experiments selected for the analysis exclusively represent warmer climates. Can the authors explain why they chose not to include model results from colder climate states (e.g., the PMIP LGM experiments)? I think the community's constraints on glacial boundary conditions is particularly strong (perhaps better than, say, the mid-Pliocene) and there is an emerging consensus that ENSO was reduced at this time (e.g., Thirumalai et al., 2024 and references therein). The authors' nonlinearity argument could be substantially bolstered by including the LGM simulation, which I expect would extend the right-hand "poleward displacement" side of the quadratic relationship in Figs. 2 and 3. But currently, without a colder state estimate, it's hard to know if the nonlinearity presented here is a universal ENSO response or if it's true only for warm states.

Unfortunately, only 5 modelling groups have uploaded LGM experiment outputs to the ESGF database (AWI-ESM-1-1-LR, CESM2-FV2, CESM2-WACCM-FV2, INM-CM4-8,

MIROC-ES2L and MPI-ESM1-2-LR, last access 31/01/2025). From those, only three models (CESM2-FV2, INM-CM4-8, MIROC-ES2L) had the minimum necessary variables (sea surface temperature and precipitation) available for download. Given that a subset of only three models is likely not representative of the LGM ensemble, we initially opted for not including LGM in our study. Nonetheless, we have now included the available LGM simulations in the analyses presented in Figure 2 and 3.

2.    A side note to the comment above: I realized mid-way through this review that I might be mistaking what "equatorward" and "poleward" migrations are in reference to. Are they in reference to the equator, or the mean position of each convection center, or something else? It may be useful to quickly define what cardinal direction the authors are referring to in the discussion (e.g., "equatorward" might be northward for the SPCZ but southward for the ITCZ, according to Fig. 4). This is important because including glacial output might instead improve the left-hand "equatorward displacement" side of the quadratic, which is currently data sparse (as I note below).

The convection-centre index is defined relatively to each model's piControl position. The origin of the graph in Fig. 2 indicates the piControl state of each model. We now included a detailed computation of convection-centres index in the main manuscript and added information in Fig. 2 for clarity (below).

To evaluate the effect of the position of the Convergence Zones on ENSO activity we develop an index that captures their combined displacement during the developing and mature ENSO phases (austral spring and summer) (Pontes et al., 2022a). The ITCZ and SPCZ positions are computed as the precipitation weighted average over latitudes in which precipitation is greater than 70% of the maximum zonally averaged precipitation in each hemisphere (0º-20ºN and 20ºS-0º, respectively). This methodology captures migrations of the ITCZ and the SPCZ independently from one another. As our objective is to quantify their overall displacement relative to the equator (e.g., equatorward shit of the ITCZ and SPCZ represent an El Niño-like mean state), we consider their absolute values. The index D is obtained through quantifying their combined displacement relative to the model's piControl position:

$$D = [|ITCZ_s| - |ITCZ_{PI}|] + [|SPCZ_s| - |SPCZ_{PI}|]$$

where the subscript 'S' indicates the position of the ITCZ and SPCZ in the perturbation scenarios and the subscript 'PI' denotes their position in their respective pre-industrial simulation. Hereafter, the index D is referred to as the convection-centers index and is, by definition, positive for poleward displacements of the Convergence Zones. It is important noting that in all scenarios the Pacific ITCZ lies in the Northern Hemisphere, therefore a poleward movement reflects a northward shift, whereas for the SPCZ, a poleward displacement reflects a southward shift.

[Figure]

Figure R2.1 – Relationship between the convection-centres index and the change in niño3 amplitude, measured by its standard deviation (s.d). The solid black line indicates the quadratic fit based on the least squares method. Banding indicates 95% confidence interval based on a 1000-sample bootstrap. The mean displacement of the convection centers boreal spring-summer is considered (i.e., encompassing developing and mature ENSO phases). R2 indicates the coefficient of determination and *a* the nonlinear coefficient of the quadratic regression model. Error estimates for R2 and alpha we calculated as one standard deviation of 1000 bootstrap realizations. The convection center index is, by definition, positive for poleward movements of the Convergence Zones referenced at each model's piControl positions.

3.      Regarding the experiments included by the authors, the $4xCO_2$ simulations does a lot of work in establishing the nonlinearity discussed in the paper. Indeed, three datapoints from this simulation anchor the left-hand "equatorward displacement" side of the quadratic function (Figs. 2b-d and 3b,c). Without these three points, I can see the remaining data forming more of a negative linear relationship, rather than a nonlinear one. Despite its importance, the $4xCO_2$ simulation results receive little attention (e.g., it is not mentioned in section 3.1 and not presented in Fig. 1). I think these results should feature at least as prominently as the others, considering their importance for the nonlinearity argument.

We thank the Reviewer for this suggestion. We have now included the 4xCO2 results in Fig. 1 and are now discussing these results in section 3.1:

In contrast, CO2-driven scenarios (SSP126-23, SSP585-23 and abrupt4xCO2) resemble an El Niño-like mean state (Figure 1d). It is interesting to note that as the SST warming intensifies, both the ITCZ and Subtropical Pacific Convergence Zone (SPCZ) tend to move equatorward (Fig. 1d-f), progressively weakening the equatorial trades and eastern upwelling. In moderate $CO_2$-driven warming scenarios (e.g., SSP126-23) these changes allow enhanced ENSO variability (Fig. 1g). Interestingly, however, ENSO variability tends to decrease under strong

warming scenarios (e.g., SSP585-23), when the ITCZ and SPCZ are nearly collapsed at the equator (Fig. 1e-g), thus resulting in a nonlinear ENSO response to the background state.

4.       Can the authors speculate on the applicability of their ITCZ response to millennial-scale climate change? I'm familiar with modeling results for the deglaciation (particularly cold stadial events) where a southward displacement of the ITCZ amplifies, rather than dampens, ENSO behavior (e.g., Liu et al., 2014; Timmerman et al., 2007). This is generally attributed to ENSO's "frequency entrainment" to the annual cycle (Chang et al., 1994, 1995; Liu et al., 2002), which is weakened when the winds shift southward. Although this mechanism is still debated, it does find support in the paleo record, where studies have noted an increase in ENSO variability from the LGM to Heinrich Stadial 1 (Leduc et al., 2009; Glaubke et al., 2024; Sadekov et al., 2013) in response to a disruption of AMOC and southward shift of the ITCZ (e.g., Mosblech et al., 2012). Is there something unique about the meltwater-induced ITCZ migrations of the deglaciation that might represent a special case to the universal mechanism proposed here? Could it have something to do with the ITCZ and SPCZ moving in one direction *together* as opposed to moving closer or farther apart, as implied in Fig. 4?

The mechanism we propose in this study should be applicable to all climate states. During North Atlantic meltwater-induced ITCZ migrations, the southward shift of the ITCZ is evident over the Atlantic and Indian basins, however the Pacific response is more complex. In the Pacific, the ITCZ migration will depend on the balance between how much intense the northeasterly trades will become due to a stronger North Atlantic subtropical circulation (Saini et al., 2025), which prevents an ITCZ southward shift, and the change in the interhemispheric temperature gradient, which tends to drive an ITCZ southward shift. It is thus possible that different intensities of meltwater pulses cause different ENSO response, as found by Glaubke et al. (2024). Modelling studies do not necessarily agree on the impact of an AMOC weakening on ENSO variability (e.g. Timmerman 2008, Oriuhuela-Pinto et al., 2022). Furthermore, our results show that the SPCZ migration is also extremely important for ENSO, a feature that has received little attention. As the reviewer mentions if the SPCZ is also moving southward, then the ENSO change will result from a delicate balance of ITCZ vs SPCZ changes.
Finally, our mechanism is in agreement with the 'ENSO's frequency entrainment' to the annual cycle. This is evident from results shown in Fig. 3a: as ENSO activity increases, so does ENSO frequency (i.e. closer to one event per year), until ENSO is fully entrained to the annual cycle (i.e. one event per year - Permanent El Niño-like mean state).
To add insights into ENSO response during climate states not covered by our analyses we have added this discussion:

      The proposed mechanism can reconcile divergences in ENSO responses in proxy-data and modelling studies. First, modelling results from the idealized abrupt4xCO2 simulations indicate inconsistent inter-model ENSO response. We showed that this is related to the sensitivity of ITCZ and SPCZ migrations in each model. This argument can be expanded for meltwater-induced AMOC weakening modelling studies and proxy-data covering Heinrich stadials and the Younger Dryas, which often do not agree on the ENSO response (Glaubke et al., 2024; Liu, 2002; Timmermann et al., 2007). This could be related to the different response of the Pacific Convergence Zones to the intensity and duration of meltwater pulses. Finally, our results provide novel insights into the relationship between ENSO frequency and the annual

cycle (Liu, 2002), suggesting that moderate equatorward shift of the Convergence Zones relaxes both mean horizontal and vertical circulations, allowing nonlinear interactions between the annual cycle and ENSO's frequency, thus increasing ENSO's frequency towards the annual cycle. Strong equatorward displacements of the Convergence Zones result in a strong annual cycle associated with an El Niño-like mean state, thus fully entraining ENSO's frequency into the annual cycle and reducing ENSO's interannual variance.

Liu, Z., 2002: A simple model study of the forced response of ENSO to an external periodic forcing. J. Climate, 15, 1088–1098.

Glaubke, R. H. et al. An Inconsistent ENSO Response to Northern Hemisphere Stadials Over the Last Deglaciation. Res. Lett. 51, (2024).

Saini, et al. Australasian hydroclimate response to the collapse of the Atlantic Meridional Overturning Circulation under pre-industrial and Last Interglacial climates. Paleoceanography and paleoclimatology (2025).

**Minor/Editorial Comments** *(Note: [] represents a deletion; [words] represents added text.)*
***Abstract***
1.      Lines 19-21: The authors may want to consider a stronger and more specific problem statement. Perhaps something like: "However, [a common mechanism that can predict ENSO variability under a range of background conditions remains elusive."]

We thank the Reviewer and have addressed this suggestion.

***Introduction***

2.      Throughout the introduction and later in the paper, the authors switch between using *ka* and *Ma* depending on the time period under discussion. For consistency, it might be worth sticking to one of these.

Given that the only period over 1Ma is the mid-Pliocene, we now use ka throughout the text.

3.      Line 62: Change "mid-Holocene" to "MH" here and throughout, as this is how you define it in the sentence prior.

This suggestion has been addressed.

4.      Line 63: These are just two of many papers you can cite here. I would either add more for completeness (e.g., for the mid-Holocene, Conroy et al., 2008; Chen et al., 2016; White et al., 2018) or cite a review like Lu et al. (2018) and the references therein.

This suggestion has been addressed.

***Data and Methods***

5.	Line 72: "...key [] periods in Earth's history. Here, we analyze [three past] climate scenarios..."

This suggestion has been addressed.

6.	Lines 72-74: You have already introduced and defined these climate periods in the introduction. You do not have to do this again here.

This suggestion has been addressed.

7.	Lines 88-89: "... removing the monthly annual cycle." Can you expand on this? What I think you're saying is that you computed the mean annual cycle at a monthly resolution and subtracted that from each monthly value, but it's not immediately clear.

Your interpretation is right. We have modified it to improve clarity:

SST anomalies are computed by removing the mean annual cycle at a monthly resolution.

8.	Line 93: "... is projected approximately 45° between..." Is this 45° longitude?

This is 45° in the EOF space (i.e. when plotting the relationship between EOF1 and EOF2). We noted that this sentence is missing a proper reference that has now been added.

Takahashi, K., Montecinos, A., Goubanova, K., & Dewitte, B. (2011). ENSO regimes: Reinterpreting the canonical and Modoki El Niño. *Geophysical Research Letters*, *38*(10). https://doi.org/10.1029/2011GL047364

9.	Line 98: "To avoid being [misled]..."

Thank you for noting this mistake.

**Results and Discussion**

10.	Line 105: I would change "Results" to "Results and Discussion"

This suggestion has been addressed.

11.	Section 3.1: I would add the $4xCO_2$ results here. I would also suggest bringing in LGM simulations (as discussed above) and include them here as well.

We have added the results from the 4xCO2 simulations here, but  we could not add results from the LGM simulations as we only have the data for three models (please see comment above). Nonetheless, the results from the LGM have been included in Figures 2 and 3.

12.	Line 131: "...ITCZ and SPCZ..." As someone who works on the deglaciation, I think about ITCZ much more and have a better grasp of how it influences ENSO (i.e., changing

the position of the trades). It might be helpful for those like me who don't think about the SPCZ much to briefly mention how it relates to ENSO. Perhaps add it to the introduction? As of right now, the SPCZ isn't mentioned until the results, so it would be nice to mention it sooner in the paper.

*This suggestion has been addressed. We have moved the first paragraph of section 3.2 to the introduction and made the necessary changes to it.*

13.     Lines 132-135: "Firstly..." Consider rephrasing. Perhaps "ENSO growth and extreme rainfall events are reduced the farther tropical convection centers are from the equator (Pontes et al., 2022). This occurs since the position of the convection centers determines the equatorial wind field and its associated upper ocean response."

*Thank you for your suggestion. It has been addressed.*

14.     Line 141: "...[which underpins] ENSO development."

*This suggestion has been addressed.*

15.     Line 151: I would delete "preliminary".

*This suggestion has been addressed.*

16.     Line 152: Add (Fig. 2) to the end of the sentence.

*This suggestion has been addressed.*

17.     Lines 155-158: Could the authors clarify what the equatorward and poleward shifts are with respect to? Is it the physical equator (0°)? The mean position of each convective center?

*We use each model's piControl position as reference. We have now included a detailed description of the calculation of the convection-centres index in main manuscript, as described in the major comment #2.*

18.     Line 163, 165, 173, etc: Consider italicizing parameters of an equation: "... ($a$ = 0.XX)..."

*This suggestion has been addressed.*

19.     Lines 182-183: Consider simplifying. "To investigate the [rainfall response to convection center migration], we..."

*This suggestion has been addressed.*

***Conclusions***

20.    Line 229-230: Eliminate redundancy. "...linking meridional shifts of the [atmospheric] convection centers, [] ocean stratification, and zonal thermocline oscillations."

This suggestion has been addressed.

21.    Line 236: "...(<9°)..." Did the authors mean <8°, as mentioned in line 157?

Thank you for noticing this typo. Yes, it should read '<8º'.

22.    Line 247-252: Is there room in an appendix of supplementary material to elaborate on these model biases and how they might influence the results? This could be helpful for non-modelers (such as myself) reading the paper. For example, I imagine the double-ITCZ problem is a relevant bias to dig into. If the SPCZ is defined as the region where precipitation is >50% the zonal average between 0° and 20°S (Line 530-531), then how might the excess precipitation south of the equator from the artificial southern ITCZ (mentioned in Line 540-541) influence that estimation?

Thank you for this suggestion. When computing our convection-centres index, an artificial ITCZ south of the equator would tend to artificially displace the position of the SPCZ equatorwards. However, we note that although it is a model deficiency, it dynamically affects the simulated ENSO response, through producing additional biases in the intensity of the trade winds and zonal SST gradient. To make the implications of this bias to ENSO clearer, we have added this discussion:

It is important noting that double-ITCZ biases may affect the calculated SPCZ position. The double-ITCZ bias is an artificial feature produced by most climate models that overestimates the tropical precipitation south of the equator in the central-eastern Pacific. This biased precipitation artificially induces the index D to capture an apparent SPCZ displaced northwards. Nevertheless, although being an artificial feature, the double-ITCZ dynamically influences the tropical Pacific climate through simulating weaker trade winds and warmer SSTs than in observations and therefore affecting the model's ENSO response. We thus do not differentiate between the southern branch of the double ITCZ and the SPCZ in our evaluation.

***Figures***

23.    Figure 1: Excellent figure. Of course, I think adding the $4xCO_2$ results and some data from a glacial simulation would make it perfect.

This suggestion has been addressed.

24.    Figure 2: Be sure to add what "equatorward" and "poleward" are in reference to. Consider also adding cardinal directions, if that helps.

We have added the definition of the convection centres index to the figure caption. Unfortunately, adding cardinal directions is not possible in this case because an equatorward shift would mean a southward shift for ITCZ and northward shift for SPCZ.

As our objective is to quantify their overall displacement relative to the equator (e.g., equatorward shit of the ITCZ and SPCZ represent an El Niño-like mean state), we consider their absolute values.

and

It is important noting that in all scenarios the Pacific ITCZ lies in the Northern Hemisphere, therefore a poleward movement reflects a northward shift, whereas for the SPCZ, a poleward displacement reflects a southward shift.

25.      Figure 3a: Consider using unique markers for each climate simulation like in Figs. 2b-d and 3b,c. It would be useful to see, for example, what proportion of the "strong equatorial shift" category of model results is from the $4xCO_2$ experiment.

Results presented in Fig. 3a are produced by a bootstrap analysis, in which we consider all simulations that lie in each group of convection-centres shift (i.e. defined by poleward, moderate equatorward and strong equatorward shifts). When applying the bootstrap analysis, a first assumption is that all these simulations are part of the same sample (i.e. same group). It is thus not possible to distinguish them after bootstrapping.

26.      Figure 4: I like this figure! I was initially confused by how it was arranged, but once I was oriented, it clicked. Maybe make the X and Y arrows a bit larger to draw the eye?

Thank you! We have made the arrows longer and wider and increased the font of the X and Y labels.

**References**
1.      Thirumalai, K. *et al*. Future increase in extreme El Niño supported by past glacial changes. *Nature* 1–7 (2024) doi:10.1038/s41586-024-07984-y.
2.      Liu, Z. *et al*. Evolution and forcing mechanisms of El Niño over the past 21,000 years. *Nature* **515**, 550–553 (2014).
3.      Timmermann, A. *et al*. The Influence of a Weakening of the Atlantic Meridional Overturning Circulation on ENSO. *J Climate* **20**, 4899–4919 (2007).
4.      Chang, P., Wang, B., Li, T. & Ji, L. Interactions between the seasonal cycle and the Southern Oscillation - Frequency entrainment and chaos in a coupled ocean-atmosphere model. *Geophys Res Lett* **21**, 2817–2820 (1994).
5.      Chang, P., Ji, L., Wang, B. & Li, T. Interactions between the Seasonal Cycle and El Niño-Southern Oscillation in an Intermediate Coupled Ocean-Atmosphere Model. *J Atmos Sci* **52**, 2353–2372 (1995).
6.      Liu, Z. A Simple Model Study of ENSO Suppression by External Periodic Forcing*. *J Climate* **15**, 1088–1098 (2002).
7.      Leduc, G., Vidal, L., Tachikawa, K. & Bard, E. ITCZ rather than ENSO signature for abrupt climate changes across the tropical Pacific? *Quaternary Res* **72**, 123–131 (2009).
8.      Glaubke, R. H. *et al*. An Inconsistent ENSO Response to Northern Hemisphere Stadials Over the Last Deglaciation. *Res. Lett.* **51**, (2024).
9.      Sadekov, A. Y. *et al*. Palaeoclimate reconstructions reveal a strong link between El Niño-Southern Oscillation and Tropical Pacific mean state. *Nat Commun* **4**, 2692 (2013).

10.     Mosblech, N. A. S. *et al.* North Atlantic forcing of Amazonian precipitation during the last ice age. *Nat Geosci* **5**, 817–820 (2012).

11.     Conroy, J. L., Overpeck, J. T., Cole, J. E., Shanahan, T. M. & Steinitz-Kannan, M. Holocene changes in eastern tropical Pacific climate inferred from a Galápagos lake sediment record. *Quaternary Sci Rev* **27**, 1166–1180 (2008).

12.     Chen, S. *et al.* A high-resolution speleothem record of western equatorial Pacific rainfall: Implications for Holocene ENSO evolution. *Earth Planet Sc Lett* **442**, 61–71 (2016).

13.     White, S. M., Ravelo, A. C. & Polissar, P. J. Dampened El Niño in the Early and Mid-Holocene Due To Insolation-Forced Warming/Deepening of the Thermocline. *Geophys Res Lett* **45**, 316–326 (2018).

14.     Lu, Z., Liu, Z., Zhu, J. & Cobb, K. M. A Review of Paleo El Niño-Southern Oscillation. *Atmosphere-basel* **9**, 130 (2018).

**Reviewer 2**

In this manuscript, the authors aim to clarify the relationship between properties of ENSO variability and the mean background climate in a range of different past and future warmer climate states. The authors employ different past climate simulations (covering the mid-Pliocene, last interglacial and mid-Holocene) and future simulations (ssp585 and abrupt 4x $CO_2$) and present their results with respect to pre-industrial climate simulations. The results show a common mechanism linking changes to the mean atmosphere-ocean circulation to an increase or decrease in ENSO variability. In fact, the authors reveal a non-linear relationship between a change in ENSO variability and the change in the meridional positions of the intertropical and South Pacific convergence zones. Three distinct regimes of ENSO variability change are identified and the processes in the tropical Pacific mean climate leading to the existence of these regimes are explained.

I believe this is a very interesting study with relevant findings. The scientific content is substantial and provides a significant contribution to the field of ENSO complexity. The analysis is sound, and the figures are very clear. I do have a few issues, related to the data selection and to the Introduction, that I would consider major (but should not require too much effort to resolve). Apart from those, I present minor comments and a few technical corrections. I recommend this manuscript to be published in Climate of the Past subject to revisions.

We thank the Reviewer for their constructive comments, which helped improve our manuscript. Below we answer the reviewer's comments and include in purple some excerpts of modified text to be included in the revised manuscript.

**Major concerns**

1.      My main issue is the following. Your aim is to analyze nonlinear ENSO sensitivity to the mean climate. I have no issues with that with regards to the past climate simulations. But the future simulations, ssp585 more so than abrupt4x$CO_2$, are transient simulations, and not in equilibrium. You seem to use data from 2015 - 2100 (if I am correct), but both the mean climate as well as ENSO characteristics change quite drastically over that period. I don't think it makes sense to take the mean climate over this whole period and to consider that ENSO characteristics are constant over this period, because they are not. Either you need to convince me that what you are doing is fine (e.g. show me that the mean climate and ENSO properties between f.e. 2015-2060 and 2061-2100 are similar enough for your analysis following some statistical test), or I would suggest you use different simulations or a different selection of years. My suggestion would be to use the simulations from the LongRunMIP project instead of the ssp585. LongRunMIP also encompasses the abrupt4x$CO_2$ simulation, so that works out well. It includes high $CO_2$ 'future' simulations, but equilibrated and not transient, so the mean climate can be considered in a mean state, and ENSO can be considered not changing over the last 100 years. Another option is to use extended SSP585 runs, but I don't think many models ran those runs, so that might give you only a small ensemble to use.

We thank the reviewer for this suggestion and agree that the ssp585 scenario during the 21st century is not appropriate for evaluating ENSO's sensitivity to the mean state. We

have removed the transient response of the ssp585 simulations (i.e. 21$^{st}$ century) from our analysis. Instead, to evaluate simulations that are closer to equilibrium we now include simulations that have extended the ssp585 and ssp126 until 2300, with atmospheric $CO_2$ concentrations fixed at their 2100 level. In the ssp585 scenario $CO_2$ emissions reduce linearly from the 2100 level to 2250, reaching a near constant radiative forcing of 12 W.m$^{-2}$ from 2250 (O'Neil et al., 2016).  In the extended ssp126 scenario the rate of negative emissions reach in 2100 is extended until 2140, stabilizing at ~2 W.m$^{-2}$ from 2200. We thus use the last 50 years of these simulations in our analyses.

We thank you for your suggestion to use LongRunMIP data, however  only annual-mean SST data is available on the LongRunMIP database, thus not allowing us to compute ENSO's metrics.  Nevertheless, LongRunMIP results described in Callahan et al. (2021) likely resemble the extended ssp585 simulations, with enhanced equatorial warming, collapse upwelling and reduced ENSO activity.

[Figure]

Figure R2.1 – Relationship between the convection-centres index and the change in niño3 amplitude, measured by its standard deviation (s.d). The solid black line indicates the quadratic fit based on the least squares method. Banding indicates 95% confidence interval based on a 1000-sample bootstrap. The mean displacement of the convection centers boreal spring-summer is considered (i.e., encompassing developing and mature ENSO phases). R2 indicates the coefficient of determination and *a* the nonlinear coefficient of the quadratic regression model. Error estimates for R2 and alpha we calculated as one standard deviation of 1000 bootstrap realizations. The convection center index is, by definition, positive for poleward movements of the Convergence Zones referenced at each model's piControl positions.

2.      I have an issue regarding reproducibility of your results because is it quite unclear what your specific data selection is. Which observational dataset is used? It is not mentioned in the main text, nor in the appendix, nor in the supplement. Do you use monthly data? And how many years? I find 100 years in "Data availability". Are the mean climate features you show based on the mean of those 100 years? For the past periods

that is fine. But most ssp585 simulations go from 2015 to 2100; does that mean you only use 85 years? The caption in Figure 1e seems to suggest so. What about the abrupt 4xCO2 simulations though? I am left to find out a lot of details about your data selection from different parts of the paper. I find that a major issue, but I think it will cost you very little time to collect those details in the Methods section.

We thank the Reviewer for noting this. For all experiments, but for ssp585 and ssp126, we use the last 100 years of each model's simulation. For the future scenarios, following your comment above, we now use the period from 2251 to 2300. Regarding the observational datasets, we use GPCP for precipitation from 1979-2022 and the HadISST for sea surface temperature over the same period. These details have now been included in the methods section:

We use each model's pre-industrial control simulation (PI; $CO_2$=284 ppmv) as reference for all scenarios. For all paleoclimate experiments (including abrupt4xCO2 and piControl) we use the last 100 years of each model's simulation. For the future scenarios, SSP5-8.5 and SSP1-2.6, we use the time slice from year 2251 to 2300 (referred to as SSP585-23 and SSP126-23, respectively) as representative of a near-equilibrium state.

and

Here, we use data from the Global Precipitation Climatology Project and the Hadley Centre Sea Surface Temperature datasets to calculate the strength of this feedback in observations.

3.      Lastly, I have some issues with the Introduction. It is clear, but incomplete (in my view). I have three recommendations:

a) First, I find it unclear from the Introduction what the research aims are. Currently, the only stated aim is in L66-67: "to investigate the effect of changes in Pacific mean state on ENSO variability." That is quite vague. Can you explain in more detail: What the main research gap is that you plan to address / what the main research aims are, what your research questions are (and why – relate this to the literature overview), and how you plan to answer those (i.e. more detail on the specific features you will analyze).

b) Too much important background information is explained after the Introduction (i.e. in the Methods and Results). For example, the different types of ENSO are only explained in the Methods – why not in the Introduction? The SPCZ is mentioned first in the Results (L118), but plays an important role throughout the paper – why it is not explained in the Introduction? The most striking is the first paragraph in section 3.2 (L129 – 147). This paragraph is a literature overview, and harbors no results, so should not be included in the Results. Please incorporate it in the Introduction.

c) The literature overview on ENSO variability in the future is incomplete. Cai et al. (2021) present a literature review on ENSO in a warm climate and show (with more confidence than the study you cite now) that ENSO amplitude increases under future warming, but also that equatorial Pacific rainfall intensifies, extreme ENSO rainfall events increase, and EP ENSO variance increases clearly while agreement on CP ENSO variance increase is low. I believe all these results resonate with your findings, so it is relevant to mention in

the Introduction. Additionally, while near-term future climate simulations (e.g. until 2100) show an intensification of ENSO variability, equilibrated long-term warming simulations show a robust decrease (Callahan et al. 2021 and Zheng et al. 2022). Both studies employ the LongRunMIP ensemble simulations, that represent equilibrated 2x, 4x, 6x and 8x CO2 increases, both abrupt and 1pct increase type simulations. Almost all these simulations show a robust decrease in ENSO variability. I find that a very significant finding to mention with regards to your study. Additionally (as mentioned in my previous point), I would suggest including these results as they provide suppressed ENSO not in the past but in a long term future, and I would be very interested to see in which of the nonlinear regimes the LongRunMIP results end up.

Cai, W., Santoso, A., Collins, M. et al. Changing El Niño–Southern Oscillation in a warming climate. Nat Rev Earth Environ 2, 628–644 (2021). https://doi.org/10.1038/s43017-021-00199-z
Callahan, C.W., Chen, C., Rugenstein, M. et al. Robust decrease in El Niño/Southern Oscillation amplitude under long-term warming. Nat. Clim. Chang. 11, 752–757 (2021). https://doi.org/10.1038/s41558-021-01099-2
Zheng, Y., Rugenstein, M., Pieper, P., Beobide-Arsuaga, G., and Baehr, J.: El Niño–Southern Oscillation (ENSO) predictability in equilibrated warmer climates, Earth Syst. Dynam., 13, 1611–1623, https://doi.org/10.5194/esd-13-1611-2022 , 2022

We thank the Reviewer for these suggestions. We agree that our introduction was incomplete. In a revised manuscript, the introduction will be significantly modified so as to clearly spell out the aims of our study, include the necessary background information and identify the literature gaps.

The explanatory paragraph of the ITCZ and SPCZ has been moved to the introduction. For reasons of shortening our manuscript, the introductory paragraph of the different ENSO types as well as their results has been moved to Appendix.

Finally, we have enlarged our literature review to better describe studies that evaluated the ENSO response in both future transient warming and long-term CO2-driven simulations.

**Minor comments**
1.      I feel that L39-40 "which has been ... instrumental period" feels out of place here. I suggest removing it here (just end the sentence) and combine it with "with strong impacts worldwide" in L36. So, end L36 at "eastern Pacific (ref)." then insert: "It has been the most important driver of year-to-year climate variability during the instrumental period, with strong impacts worldwide."

Due to the need for shortening our main text, we have removed this sentence.

Subscripts and superscripts need to be included consistently. CO2 should be $CO_2$ in L60, L73, L74, L76 and throughout. You write 21st, 4th and 6th in L66-68, but 21st in L75. Choose a style and apply consistently. In L152 R2 should be $R^2$.

Thanks for your careful reading. Subscripts and superscripts have now been carefully checked.

2.    L60 "both interglacials" – I understand you mean LIG and MH. However, the mid-Pliocene simulations you use from PlioMIP2 are also an interglacial period. I feel that is a detail worth mentioning (maybe in the Methods where you introduce PlioMIP2, explain that the simulation protocol is tuned to the KM5c interglacial).

We have replaced the word 'interglacials' in this sentence by 'LIG and MH'. In addition, we now mention that PlioMIP2 targeted the KM5c interglacial.

3.    L80-82: I think you need to rewrite this sentence. C-index is an index that represents the CP-ENSO variability, but now it reads as if C-index is the name of a type of ENSO variability (which is, in fact, CP ENSO).

We agree with the reviewer. This sentence was rephrased to:
To account for ENSO complexity, we evaluate the central (CP) and eastern Pacific (EP) ENSO types (Takahashi et al., 2011)

For reasons of shortening the manuscript the analysis of the C- and E-indices has now been moved to Appendix A, where we introduce the C and E indices:
As such, indices for CP-ENSO (C-index) and EP-ENSO (E-index) were computed through combining the first two Empirical Orthogonal Functions of SST anomalies in tropical Pacific (15ºS-15ºN; 140ºE-80ºW).

4.    L91: only the ssp585 data is detrended? I would say, for consistency, to apply detrending to all simulations (although it will likely not affect the results substantially).

We agree that other simulations may not be at equilibrium or present possible model drift. To account for this, we have now removed the linear trend from all simulations.

5.    L103-105 I would like to see included in the text how many models you end up using per scenario/past period after the selection. From the supplement it seems that you end up with quite some models for the future simulations, but only a relatively small selectionfor the past periods. I find that relevant information to include here.

We have addressed this suggestion.

6.    L112 WES feedback could merit a citation

We now cite the study from Vimont et. al. 2001.

Vimont, D. J., Battisti, D. S. & Hirst, A. C. Footprinting: a seasonal connection between the tropics and mid-latitudes. *Geophys. Res. Lett.***28**, 3923–3926 (2001).

7.    L113 "the increased difference in inter-hemispheric warming" -- this is not something

that is necessarily shown by your results, is it? the SSTs in a-c mainly show a reduced meridional gradient. What I think you mean is the heat asymmetry between both hemispheres, correct? I know from your previous work on the Pliocene (Pontes et al 2022) that this heat asymmetry, stemming from planetary albedo differences related tovegetation, ice sheets, or orbital differences, drives the northward shift of the ITCZ. For the LIG and MH, I am left to deduce the cause of this inter-hemispheric difference from a sentence in the Introduction (L60-62): "higher boreal summer insolation", which causes a difference in interhemispheric warming. This is a lot of deduction work for a reader, so I think this merits a sentence here explaining in more detail what is meant by 'increased difference in interhemispheric warming' and clarify this is based on previous findings and not on results in this work.

We agree with the Reviewer that we do not explicitly show the causes of increased inter-hemispheric temperature gradient in past periods. We have now modified this sentence (see below) and cited the appropriate studies.

In the mid-Pliocene simulations, this is attributed to enhanced albedo feedback in the Northern Hemisphere due to large sea-ice and land ice losses and enhanced vegetation area, while increased insolation in the Northern Hemisphere is the main driver in the MH and LIG (Brierley et al., 2020; Otto-Bliesner et al., 2021; Pontes et al., 2020).

8.     L118 regarding 'equatorward' and relating to the other reviewer's comments: to me it is quite clear that the ITCZ in the Pacific is always located in the Northern Hemisphere, so I understand your usage of equatorward and poleward here. However, that might not be obvious to all reader, so maybe you can clarify this point.

We have now added a clarification to this point:
It is important noting that in all scenarios the Pacific ITCZ lies in the Northern Hemisphere, therefore a poleward movement reflects a northward shift, whereas for the SPCZ, a poleward displacement reflects a southward shift.

9.     L124-126 "In this context ... mean climate" -- I am not convinced by this formulation. Ithink there are two different formulations in this sentence, one more related to physics(which I suggest keeping), and the latter more related to how ENSO is defined. SinceENSO is defined as an anomaly to a mean climate, it indeed would follow that when amean climate is different, ENSO will be different. But that to me feels quite arbitrary and does not reveal any new information on the physical link between the mean state and ENSO variability, which is what you are trying to establish in this study. The first part is more interesting: the strength of the mean atmosphere-ocean circulation can modulate ENSO variability. If possible, I would expand more on that, e.g. a strong mean circulation (La Niña like mean state) inhibits the development of anomalies, effectively suppressing ENSO variability, while a weak mean circulation (El Niño like mean state) is more favourable for the development of anomalies that can lead to ENSO events.

We thank the Reviewer for this suggestion. We thus have modified the sentence to:

These studies usually suggest that changes in the equatorial Pacific mean state (e.g., upwelling, zonal SST gradient and ITCZ) are responsible for changing ENSO activity, thus contributing to

the hypothesis that ENSO is modulated by the background state. Under this hypothesis, a strong mean circulation (i.e. La Niña-like mean state) inhibits the development of wind and SST anomalies, effectively suppressing ENSO variability, while a weak mean circulation (i.e. El Niño like mean state) favors the development of anomalies that can lead to ENSO events.

10. L131-132 "whose meridional positions affect ENSO dynamics" – I think this statement merits a reference

We now cite the study from Hu and Fedorov 2018, and Brown et al. 2020.

Hu, S., & Fedorov, A. v. (2018). Cross-equatorial winds control El Niño diversity and change. In *Nature Climate Change* (Vol. 8, Issue 9, pp. 798–802). Nature Publishing Group. https://doi.org/10.1038/s41558-018-0248-0

Brown, J. R., Lengaigne, M., Lintner, B. R., Widlansky, M. J., van der Wiel, K., Dutheil, C., Linsley, B. K., Matthews, A. J., & Renwick, J. (2020). South Pacific Convergence Zone dynamics, variability and impacts in a changing climate. *Nature Reviews Earth & Environment*, 1–14. https://doi.org/10.1038/s43017-020-0078-2

11. L151-152 "the ENSO-convection centers relationship" – propose to change this to "the relationship between ENSO amplitude change and the displacement of the convection centers"

This suggestion has been addressed.

12. L160 (regarding the previous paragraph): I agree with the other reviewer that it seems like the abrupt4xCO2 results are important for establishing a quadratic relationship over a linear one. However, I think you already showcase that even without the abrupt 4xCO2 simulations, a quadratic fit is better than a linear fit for all ENSO indices (namely in supplement figure S4). It would be worth it to mention this result in the main text.

We have now added to our set of scenarios the extended simulations of the ssp585 scenario until 2300 (Fig R2.1) that adds more confidence to results indicated by the abrupt4xCO2 simulations. As such, both ssp585-23 and abrupt4xCO2 results are now thoroughly described throughout the manuscript and their mean state changes explicitly shown in the revised Fig 1.

13. L161-L165: Are these findings based on the results from all models or only the selected subset? Do these results relate to the curves shown in Figure 2? Please clarify

Yes, these findings are based on the results shown in Fig. 2, considering the subset of models only. We now cite '(Fig. 2)' in these sentences to avoid any misunderstanding.

14. L163-165: the description of the quadratic model could be moved to the Methods

This suggestion has been addressed.

15.   L165-L169: you use 10 models for the ssp585 scenarios. is the difference between 6/10 agreeing or 7/10 agreeing what you call 'non robust increase' and 'significantly increased' ..? also, I am confused where you state 'in our subset of models' and in the next sentence 'if selecting models based on …'. Isn't this subset you mean already only those models that pass this criteria test?

When applying both SST advection and convective feedback criteria we end up with 10 models for the ssp585, and when applying the SST advection criteria only, we end up with 18 models. As such, the difference we were referring to is from 6/10 to 13/18 models. However, following your major recommendation we have now opted to use simulations that are closer to equilibrium than the transient ones. As such, we have excluded the transient response of the ssp585 scenario from our main analyses. Analysis of the central eastern Pacific types of ENSO were also moved to Appendix, where we also show results for the ssp370 scenario for completeness.

16.   L195 "SST variability" I think you mean that there is a small model spread in the mean SST in the Niño3 region. A small range in SST variability would suggest that most models agree on the amplitude of ENSO, which is not what you seem to imply.

We did mean 'SST variability' because Fig. 3a is showing the interannual SST variability during a 100-year period, and not each model's results. In this bootstrap analysis we consider that all models that lie in the different groups (defined by the convection-centres index) are part of the same sample (i.e. representing the same state of ENSO activity). Thus, after bootstrapping it is not possible to distinguish between model and thus show the first 100 years sample of the bootstrap analysis.

17.   L213 (regarding previous paragraph): following these results, I would be interested to see the ENSO amplitude change as a function of the change in easterlies variability. It would probably show a linear relationship where a weaker (stronger) ENSO relates to a weaker (stronger) easterlies variability. However, it would not show that the weaker ENSO/weaker easterlies variability is two different regimes: where the convection centers are shifted poleward or strongly equatorwards. Your results and conclusions could harbour a recommendation to the ENSO research community to not just look at this ENSO change - easterlies variability relationship, but also investigate the shift in convection centers, if one truly wants to know which regime in ENSO suppression is occurring. I think that is a valuable recommendation to state here on in the Conclusions).

We thank the Reviewer for this suggestion. We have plotted the relationship between niño3 amplitude change and the changes in easterly wind variability over the western equatorial Pacific. It shows a linear relationship (R2 = 0.67 and r (Pearson's) = 0.8). Furthermore, it groups together CO2-driven and paleo simulations, which have a very distinct mean state, in the reduced ENSO variability and easterlies variability quadrant. This analysis is thus not helping in explaining the mechanisms of the nonlinear ENSO sensitivity to the mean state. We have incorporated Fig. R2.1 below in Fig. 3 of the main manuscript, added a discussion in section 3.2 and a recommendation to research community in the 'Conclusions':

Section 3.2: Despite the consistence between changes in easterly wind variability and the position of the Convergence Zones, changes in wind variability cannot solely explain the nonlinear sensitivity of ENSO across climate states (Fig. 3b). The relationship between changes in easterly wind variability and ENSO is linear and does not distinguish between mean states of reduced (e.g., paleo) and amplified (e.g., future) equatorial warming.

Conclusion: Our findings suggest that meridional shifts of the Convergence Zones modulate key processes for ENSO development, such as equatorial convection, trade winds variability and zonal thermocline oscillations.

[Figure]

Figure R2.2 – Relationship between change in niño3 amplitude and the change in easterly wind variability, as measured by their standard deviation (s.d). Coefficient of determination ($R^2$): 0.67±0.01 and Pearson's correlation coefficient (r): 0.8. Solid line indicates the linear regression between the two samples.

18.  L231 "three key background conditions": but these are background states with regards to a PI control, so I would suggest calling it something like 'anomalous background states', or 'three distinct regimes deviating from the PI control'.

This suggestion has been addressed. We have modified the sentence to:

The interplay of these processes results in a nonlinear ENSO sensitivity to mean states, encompassing three distinct background conditions deviating from the pre-industrial climate (Fig. 4)

19.  L231 related to this, you follow up with mentioning these three regimes. For clarity for the reader, I would suggest numbering these regimes, e.g. in L232 1) A poleward migration ..., L236 2) A moderate equatorward shift... and L239 3) A strong equatorward shift (remove "finally")

This suggestion has been addressed.

20.   L238: "...dynamical coupling." – I would add ", and thus dampened ENSO variability." To that sentence for clarification.

This suggestion has been addressed.

21.   L249-250: "Nonetheless, ... selection criteria." – I find this a strange read, and it should/could include a suggestion for the community. Propose to rewrite: "In this study, we applied two model selection criteria to reduce the influence of such biases, and we recommend others to employ similar model selection steps when investigation ENSO complexity."

Thank you for this suggestion. It has been addressed.

22.   L261-263: "This may ... biodiversity loss." – I find this a highly suggestive sentence which seems to be included to increase the relevance of this study. I would suggest removing it. The study is relevant in its own right. I would recommend including a different take-away (more related to the study results), or just remove this sentence altogether.

This suggestion has been addressed. We simply removed the sentence due to the need for shortening the manuscript.

Figures:
23.   Figure 1: Am I correct in assuming panel e shows all model simulations, and not just the models selected based on the criteria mentioned in the methods? I think it makes sense to include all models here, since you are just looking at mean climate and ENSO and notyet the relationship between the two. However, I would clarify in the Methods that you include all models for this result, and only move forward with a smaller subset for the later results.

You're right. This figure includes all models, without yet applying the selection criteria. This has been clarified in the figure caption:

Analyses shown in this figure include results from all models, with no selection criteria being applied.

24.   Figure 1: you might want to extend the arrows with the correct units to all panels (a-d) so that it is clear from the figure itself which panels show wind stress, and which show surface wind. Now, from the panels itself it is not clear (although the caption clarifies).

This suggestion has been addressed.

25.   Figure 2a: Is it really necessary to include this figure with the Niño regions..? Consider leaving it out.

Thanks for your suggestion. We have removed Fig. 2a.

26.    Figure 3a: am I correct in assuming that 'control' here means PI control data, and all the other data is the grouped results of all the past and future simulations? if yes, are all model simulations included, or only the subset of models that pass the criteria test..? Please clarify

Yes, your interpretation is correct. We have replaced 'control' by 'piControl' in this figure. Most of the simulations in the 'poleward shift' group are paleo simulations, and most of the simulations in the 'moderate' and 'strong equatorward shift' groups are future scenarios, including the abrupt4xCO2. As the main objective of this analysis is to elucidate the mechanisms responsible for driving changes in ENSO variability, only the subset of models is used. This has been clarified in the figure caption.

Results shown in this figure were obtained through analyses of the subset models that best simulate ENSO nonlinear processes (Supplementary Information Table S1).

27.    Figure 3a: 'strong equatorward' is defined as >9deg. looking at fig 3b, to me that seems like only 1 simulation belonging to abrupt4xCO2. Is that correct? Because that would imply that your findings on the strong equatorward rainfall-SST relationship is based on 1 model and 1 simulation, making it hard - I think - to connect meaningful conclusions to that result.

Yes, in the previous version of our manuscript, the 'strong equatorward shift' group included a single simulation, which was the model CESM2- under the abruptx4CO2 scenario. However, following your comment #1, we now include results from the ssp585 scenario between 2251 and 2250. Furthermore, we have relaxed our model selection criteria to select models that realistic simulate the nonlinear convective feedback only. This is the most important feedback in evaluating the relationship between ENSO variability and the position of the Convergence Zones. The inclusion of these models in our analysis has increased the number of simulations in the 'strong equatorward' group to 8.

28.    Figure 3c: if I am correct, the triangles and squares in panel c are the wrong way around. Please change accordingly

Thank you for noting this. It has been corrected.

29.    Figure 4: I would suggest naming these panels (a), (b) and (c).

This suggestion has been addressed.

**Technical corrections**

We thank the Reviewer for noting these technical corrections. As the text has been modified, the location of these corrections has changed, but we carefully checked our manuscript to address them.

L29: El Niños → El Niño events

L31: how tropical → how the tropical

L41: likely the → likely that the

L61-62 "in the Northern Hemisphere" can be removed ("boreal" already implies this)

L66: in Pacific → in the Pacific

L130: position → positions

L141: coupling, important → coupling, which is important

L205: In this context, we found → We find

L228: The results described above → Our results

L236: Moderate → A moderate

L274: Simulation → Simulations

L281: the pre-industrial climate → pre-industrial climate simulations

**Reviewer 3**

This study combines state-of-the-art model simulations of past climates and future warming to evaluate ENSO activity throughout a wide range of climate states. The authors find that the sensitivity of ENSO to the background climate is nonlinear and tied to the climatological position of the tropical Pacific convection centers, the ITCZ and SPCZ. Results of this study provide a comprehensive mechanism of how tropical Pacific mean state modulates ENSO activity.

I find this is a novel and interesting study and the analysis is basically sound. Nevertheless, as previously mentioned by the other two reviewers, the writing can be further refined so that the paper will be more readable.

Please see my specific comments below.

We thank the Reviewer for their constructive comments, which helped improve our manuscript. Below we answer the reviewer's comments and include in purple some excerpts of modified text to be included in the revised manuscript.

1.      As seen in the dispersion diagrams and discussed in the manuscript, abrupt 4xCO2 simulation results show strong equatorward shift of convection centers, which is distinct from the past climates. I was wondering how the location of convection centers and ENSO-convection centers relationship change in other less aggressive CO2 emission scenarios. It will be helpful to show these results since "across climate states" is mentioned in the title of this study.

We thank the Reviewer for this suggestion. We have added data from simulations ssp126 l averaged over years 2251-2300, when the climate is closer to an equilibrium state. For completeness, we have also added results from the ssp370 scenario ("most likely scenario") in Appendix, averaged over years 2051-2100. These simulations show that ENSO variability significantly increases (95% level) in both scenarios, though with larger spread in ssp370.

[Figure]

Figure R2.1 – Relationship between the convection-centres index and the change in niño3 amplitude, measured by its standard deviation (s.d). The solid black line indicates the quadratic fit based on the

least squares method. Banding indicates 95% confidence interval based on a 1000-sample bootstrap. The mean displacement of the convection centers boreal spring-summer is considered (i.e., encompassing developing and mature ENSO phases). $R^2$ indicates the coefficient of determination and *a,* the nonlinear coefficient of the quadratic regression model. Error estimates for $R^2$ and alpha we calculated as one standard deviation of 1000 bootstrap realizations. The convection center index is, by definition, positive for poleward movements of the Convergence Zones referenced at the model's piControl position.

2.      It's interesting to see that the left branch of the quadratic regression curve is due to mid-Pliocene scatters in Fig. 3c. In contrast, there are scatters mainly from the abrupt 4xCO2 simulations at the left branch of the quadratic regression curve in Fig. 3b. Could you provide some explanation/discussion on the reason of this? In addition, why there are not many abrupt 4xCO2 scatters in Fig. 3c?

We thank the Reviewer for noticing this. This was a typo.  Symbols for 4xCO2 and Pliocene simulation were reversed in this figure. This has now been corrected.

3.      Fig. 1: It would be helpful if the color bars in Fig. 1a-1d could be revised to make positive values in warm colors (yellow and red) and negative values in cold colors (blue).

We opted for centering the warm and cold colors in the average warming over the tropical Pacific because cold colors align with regions of stronger winds and vice-versa, in agreement with changes expected from the wind-evaporation-SST feedback. Thus, representing how wind intensity affects the zonal SST gradient in the equatorial Pacific. Also, as nearly all the simulations used in our study represent warmer climates than the pre-industrial control. Therefore, most of the changes in SST are positive and thus fewer colors (yellow and red) would not highlight the patterns.

**Technical corrections:**

We thank the Reviewer for noting these technical corrections. As the text has been modified, the location of these corrections has changed, but we carefully checked our manuscript to address them.

Line 30: Central El Niño events -> Central Pacific El Niño events
Line 58: ~129 116 thousand years ago -> ~129-116 thousand years ago
Line 77: CO2to -> CO2 to
Line 167: Figure 2a -> Figure 2b
Line 176: Figure 2b -> Figure 2d
Line 477: boreal spring-summer -> austral spring-summer